# Analysis of Volatile Aroma Components and Regulatory Genes in Different Kinds and Development Stages of Pepper Fruits Based on Non-Targeted Metabolome Combined with Transcriptome

**DOI:** 10.3390/ijms24097901

**Published:** 2023-04-26

**Authors:** Chuang Huang, Peixia Sun, Shuang Yu, Genying Fu, Qin Deng, Zhiwei Wang, Shanhan Cheng

**Affiliations:** 1Sanya Nanfan Research Institute of Hainan University, Hainan Yazhou Bay Seed Laboratory, Sanya 572025, China; 2Key Laboratory for Quality Regulation of Tropical Horticultural Crops of Hainan Province, School of Horticulture, Hainan University, Haikou 570228, China

**Keywords:** pepper, volatile aroma compounds, non-targeted metabolome, transcriptome, co-expression network

## Abstract

Aroma is a crucial attribute affecting the quality of pepper and its processed products, which has significant commercial value. However, little is known about the composition of volatile aroma compounds (VACs) in pepper fruits and their potential molecular regulatory mechanisms. In this study, HS-SPME-GC-MS combined with transcriptome sequencing is used to analyze the composition and formation mechanism of VACs in different kinds and development stages of pepper fruits. The results showed that 149 VACs, such as esters, alcohols, aldehydes, and terpenoids, were identified from 4 varieties and 3 development stages, and there were significant quantitative differences among different samples. Volatile esters were the most important aroma components in pepper fruits. PCA analysis showed that pepper fruits of different developmental stages had significantly different marker aroma compounds, which may be an important provider of pepper’s characteristic aroma. Transcriptome analysis showed that many differential genes (DEGs) were enriched in the metabolic pathways related to the synthesis of VACs, such as fatty acids, amino acids, MVA, and MEP in pepper fruits. In addition, we identified a large number of differential transcription factors (TFs) that may regulate the synthesis of VACs. Combined analysis of differential aroma metabolites and DEGs identified two co-expression network modules highly correlated with the relative content of VACs in pepper fruit. This study confirmed the basic information on the changes of VACs in the fruits of several Chinese spicy peppers at different stages of development, screened out the characteristic aroma components of different varieties, and revealed the molecular mechanism of aroma formation, providing a valuable reference for the quality breeding of pepper.

## 1. Introduction

Peppers are one of the most important vegetables and originate from the tropical and subtropical regions of Central and South America, where more than thirty closely-related species of the genus Capsicum are found. There are six economically cultivated species: *Capsicum chinense*, *C. annuum*, *C. frutescens*, *C. baccatum*, *C. assamicum*, and *C. pubescens* [1], with different fruit shapes, sizes, spiciness, aromas, and tastes. Pepper fruits are rich in capsaicin, minerals, vitamin C, etc., with properties such as nutritional supplement, antioxidant, antibacterial, anti-inflammatory, thrombus dissolution, blood pressure reduction, and cholesterol reduction. As a result, they are widely used as an application in medical, antiseptic, chemical, food, and other fields [2,3].

Fruit aroma, determined by the composition and content of volatile flavor compounds, is a crucial indicator for evaluating fruit quality, guides consumer choices in many cases [4], and is closely related to human health and nutrition [5]. About 2000 volatile aroma compounds (VACs) have been identified in different species of plants, including esters, aldehydes, alcohols, phenols, terpenes, etc., which are of great significance for the complete aroma characteristics of fruits [6,7]. As the most abundant secondary metabolites in plants and important contributors to the aroma of fruits, volatile terpenoids (including sesquiterpenoids, hemiterpenoids, monoterpenoids, and diterpenoids) are synthesized mainly through the mevalonic acid (MVA) and methylerythritol phosphate (MEP) pathway, and terpene synthase (TPS) is the most critical enzyme for the final synthesis of terpenes [8,9]. The second largest category of VACs in plants are volatile benzenoid compounds, such as phenylacetaldehyde, methyl salicylate, benzyl benzoate, eugenol, and isoeugenol, which mainly come from aromatic amino acids (L-phenylalanine) and are then synthesized by the shikimate/phenylalanine biosynthetic pathways [10,11]. Most of the volatile straight-chain aldehydes, alcohols, and esters in aroma components are synthesized using unsaturated fatty acids (linolenic acid and linoleic acid) as precursors through the lipoxygenase (LOX) pathway. Linoleic acid and linolenic acid are converted into the hydroperoxide intermediate catalyzed by LOX and then form corresponding straight-chain aldehydes, alcohols, and esters under the catalysis of hydroperoxide lyase (HPL), alcohol dehydrogenase (ADH), and alcohol acyltransferase (AAT) [12,13]. In many flowers and fruits, valine, leucine, and isoleucine precursors can also produce VACs, such as branched-chain aldehydes, alcohols, and esters through the branched-chain amino acid pathway (BCAAP), in which branched-chain aminotransferase (BCAT) is the key enzyme [14,15,16]. In addition, many studies have shown that the same VACs can be regulated by multiple genes, and the same gene can regulate the synthesis of multiple VACs [17]. Meanwhile, the non-canonical biosynthesis pathways of VACs exist in plants [18], and many TFs are involved in regulating the formation of aroma substances [19,20]. Therefore, the biosynthesis of VACs is a very complex process involving multiple metabolic pathways and gene expression regulation.

Fruit aroma is an important quality component and processing demand of pepper. Therefore, as early as the 1960s, 60 VACs have been identified in Californian green bell peppers and 2-methoxy-3-isobutylpyrazine, methyl salicylate, limonene, linalool has been identified as the main aroma component [21]. Thereafter, 102, 136, and 63 VACs were identified at the two ripening stages of *Capsicum chinense* fruits, including Yucatan peppers, Cachucha peppers, and CNPH4080 peppers, where hexyl isovalerate, hexyl dimethyl butyrate, and hexyl valerate were the main components [22,23]. In Zunyi chili pepper (*Capsicum frutescens*), among the 85 identified VACs, 40 components appeared at the green stage, 53 at the breaking stage, and 65 at the red color stage. With fruit ripening, the proportion of ketones, alkenes, and aldehydes increased significantly, while the proportion of alcohols and esters decreased significantly [24]. 

According to the published literature, the research on VACs in pepper has only been conducted in a few cultivated species or varieties, and the research results have revealed that there is significant genetic diversity, geographical uniqueness, and developmental stage differences in VACs in pepper fruits [21,22,23,24]. Although it is not the origin of peppers, China has collected abundant chili resources and selected tens of thousands of varieties during long-term cultivation. The aroma information on these resources and varieties still lacks extensive research, especially around the aroma information of a batch of local characteristic spicy peppers. For example, the Hainan Huangdenglong pepper, as a local characteristic variety, has processed products that are popular worldwide, but the formation and change of VACs in its fruits are still unclear, leading to confusion among market products. More importantly, compared with the in-depth research on the synthetic pathway and molecular regulation mechanism of aromatic compounds in passionfruit, apple, green tea, and other horticultural products [25,26,27,28,29], the research on the molecular mechanism and regulation of the formation and change of VACs in pepper is still blank.

In this paper, the composition and relative content of VACs in pepper fruits from different varieties and developmental stages were determined by headspace solid-phase microextraction (HS-SPME) combined with gas chromatography–mass spectrometry (GC-MS). Moreover, RNA sequencing (RNA-seq) technology was used to analyze the gene expression of fruits in corresponding varieties and development stages, and the structural genes and co-expression networks related to the synthesis path of aroma volatiles were identified by metabolome–transcriptome association analysis. The results will be helpful in explaining the reasons for the different aromas of pepper fruits from different varieties and developmental stages and analyzing the metabolic pathways and molecular regulation mechanisms of VACs in pepper fruits from the omics level, which will lay a foundation for further improving the aroma of pepper fruits and promoting the improvement of pepper fruit flavor through gene and metabolic engineering methods.

## 2. Results

### 2.1. Volatile Aroma Characteristics of Pepper Fruit from Different Samples

The specific sample phenotype information is shown in Figure 1 and Appendix A, the total ion chromatograms for the GC-MS detection are shown in Appendix A, and the QC sample quality test is shown in Appendix A. A total of 149 VACs were identified in 36 samples (four varieties × three developmental stages × three biological replicates) and were composed of 53 esters (35.57% of the total), 27 aldehydes (18.12%), 10 alcohols, 20 hydrocarbons (13.42%), 20 terpenoids (13.42%), and 19 other VACs (Table 1). As the dominant volatiles, esters mainly include Z17 (trans-2-hexenyl isovalerate), Z11 (methyl salicylate), Z7 (n-amyl isovalerate), Z29 (6-methylhept-4-en-1-yl 3-methyl-butanoate), Z12 (4-methylpentyl 2-methylbutanoate), Z28 (6-Methyl-4-heptenyl 2-methylbutanoate), Z39 (4-methylpentyl 8-methylnonanoate) components, followed by aldehydes such as Q3 (Hexanal), Q4 (e-2-hexenal), Q16 (9-undecenal, 2,10-dimethyl-), Q21 (pentadecanal), etc. The third largest number are terpenes and hydrocarbons, the former mainly including T3 (Z-β-Farnesene), T4 (himach-ala-2,4-diene), T1 (α-Cubenene), T20 (α-Ionone), the latter comprising TH15 (hexade-cane), TH4 (2-methyl-hexadecane), TH12 (tetradecane), etc. In this experiment, only three ketones and four phenolic components were detected.

We found that the types of volatile compounds detected in different samples were basically the same, while the total relative content of volatiles showed obvious recognizable differences (Figure 2a and Table 1). With the development of fruits, the relative content of total volatiles in GJ and HDL1 gradually increased from 663.67 µg/kg and 811.76 µg/kg to 931.69 µg/kg and 1210.62 µg/kg, respectively. CTJ showed a trend of increasing first and then decreasing, with a maximum value of 761.66 µg/kg, while HDL2 showed a trend of decreasing from 1082.97 µg/kg at the green stage to 796.88 µg/kg at the mature stage. In addition, the concentration of total VACs compounds between different varieties at the same stage showed noticeable differences, and the concentration of total VACs in HDL2 (1082.97 µg/kg) was significantly higher than that of other varieties at the green stage, while HDL1 was significantly higher than other varieties at the breaking stage (1070.67 μg/kg) and mature stage (1210.62 μg/kg).

The relative content of VACs (esters, aldehydes, alcohols, terpenoids, etc.) from different samples exhibited a dynamic change in different groups (Figure 2b and Appendix A). In the four varieties, the content ratio of volatile esters in HDL1 and HDL2 during the whole stage reached 50–60%, which was much higher than that of other groups. The relative content of esters in GJ in the green stage was the highest (56.62%), while the proportion of esters in the breaking and maturation stages decreased slightly. Although the proportion of total volatile esters in CTJ was relatively low, it was still above 30.64%. The aldehydes accounted for 16.33% (HDL1), 15.96% (HDL2), 22.84% (GJ), and 24.54% (CTJ) at the green stage, respectively, and decreased gradually with the development of the fruit. The proportion of alcohols in CTJ, HDL2, and GJ increased with fruit development, from 5.41%, 4.43%, and 5.54% to 18.24%, 10.41%, and 9.47%, respectively, while HDL1 showed the opposite accumulation pattern, with 16.33% at the green stage to 3.17% at the maturation stage. Besides CTJ, terpenoids had a high content in at least one developmental stage in the other three pepper varieties. The proportion of terpenoids in GJ increased significantly at the breaking and maturation stages (about 20%), which became the second only to esters in pepper fruit. However, there was an opposite phenomenon in HDL2. Terpenoids were mainly accumulated at the green stage (15.12%) and significantly decreased to 2.18% at the maturation stage of HDL2. In CTJ and HDL1, the relative content of terpenoids maintained a stable proportion and was no significant change at three stages, ranging from 4.64 to 6.01% (CTJ) and 11.73 to 14.69% (HDL1), respectively. In addition, the total volatile esters content in HDL1 (712.61 µg/kg) was significantly higher than that in other varieties at maturity, while aldehydes (38.42 µg/kg) and alcohols (47.00 µg/kg) were significantly lower than other varieties (Appendix A).

### 2.2. Analysis of Differential VACs in Pepper Fruits from Different Samples

The difference in VACs from different pepper fruits was analyzed using the relative content of each of the metabolites. Principal component analysis (PCA) showed that the samples were significantly different (Figure 3a). Hierarchical cluster analysis (HCA) indicated that the experimental data were reliable (Figure 3b). In the cluster heat map (Figure 3c), the VACs of different pepper fruit showed significant differences. Most of them were significantly accumulated in the middle and late stages (breaking and maturation stages), clearly separated from the green stage, while metabolites of specific biochemical pathways were without obvious clustering. Moreover, the accumulation patterns of VACs were different between different varieties at the same stage. The differential accumulation of these volatile aroma substances may be the reason for elucidating the aroma formation mechanism of pepper fruit.

Moreover, we constructed PLS-DA models of VACs to analyze the similarities and differences of VACs in pepper fruits at different development stages, and in different varieties of pepper fruits at the mature stage, based on the fact that fresh peppers are mostly eaten at the mature stage. The quality test results showed that the PLS-DA model was reliable (Appendix A). Finally, according to the screening conditions VIP > 1, *p* < 0.05, and |log2FC| ≥ 1, 70 significantly different metabolites (Appendix A) from all the detected VACs were screened. There were 24 different VACs at three development stages fruits of CTJ (CTJ-a vs. CTJ-b and CTJ-a vs. CTJ-c), among which 10 VACs were significantly up-regulated at the breaking and maturation stages compared with the green stage 30 different VACs in GJ (GJ-a vs. GJ-b and GJ-a vs. GJ-c), 12 VACs were significantly up-regulated and 9 VACs were significantly down-regulated at the breaking and maturation stages compared with the green period, Among them, up-regulated VACs included Z8 (4-Methyl-pentyl isobutyrate), Z28, TH4,TH12, T20, and the down-regulated VACs mainly included O4 (2-pentyl-furan), C8 (e-2-hexadecacen-1-ol), Q3, Q4, etc. 32 significantly different VACs were found in the HDL1 comparison group at different developmental stages (HDL1-a vs. HDL1-b and HDL1-a-V HDL1-c), 12 VACs such as Z7, Z29, Z31 (valeric acid, benzyl ester), and TH5 (heptane) were significantly upregulated compared to the green stage in the middle and late stages, while Z11, Z12, Z15 (butyric acid, 2-methyl, hexyl ester), C3, C8, and the other 9 VACs were significantly downregulated. In addition, 30 VACs were identified as significant differences in HDL2 (HDL2-a vs. HDL2-b and HDL2-a vs. HDL2-c), including 6 up-regulated and 12 down-regulated VACs in the middle and late stages. Among them, up-regulated VACs included Z39, Z22 (4-methyl-amyl tiglate), C9, TH19 (z-3-heptadecene), etc., and down-regulated VACs mainly included Z35 (Isopentyl 8-methylnon-6-enoate), Z36 (pentadecanoic acid, 3-methylbutyl ester), Z23 (5-methyl-hexyl 3-methylbutanoate), Z11, Z12, etc. Forty VACs showed significant differences in mature pepper fruits of different varieties (GJ-c vs. CTJ-c, HDL1-c vs. CTJ-c, and HDL2-c vs. CTJ-c), among which volatile esters, aldehydes, alcohols, terpenoids, and hydrocarbons were the main VACs, such as Z12, Z13, Q25 (z-9,17-octadecadienal), C8, T3, Q16, etc. The VACs in pepper fruits of different varieties showed different accumulation patterns compared with CTJ mature fruits, HDL1 and GJ had more up-regulated metabolites, while HDL2 had more down-regulated metabolites. Notably, 31 of all 53 esters (58.49% of the total) showed significant differences in different comparison groups, while 22 esters were branched-chain esters, indicating that esters (especially branched-chain esters) may play a more important role in the change of pepper fruit aroma.

### 2.3. Screening of Marker Aroma Compounds in Pepper Fruits from Different Samples

Interestingly, most of VACs with high content at the green stage decreased gradually with fruit ripening, significantly different from the differential metabolites with high abundance in chili fruit at the breaking and maturation stages. Therefore, to screen out the marker aroma compounds in pepper fruits at different stages, we performed PCA analysis on the relative expression of 70 selected differential metabolites. In the PCA loading plot (Appendix A), the variable metabolites in different quadrants were the most important VACs describing the specific stage of pepper in this study. The results showed that Q3 and Z11 were the marker volatiles of CTJ at the green stage, Z13 and TH13 (tetradecane, 2-methyl-) were the typical aroma components of CTJ at the breaking stage, and F1 and Z12 were the marker aroma compounds at the mature stage. In GJ, Z11, Z12, and Q3 were significantly expressed at the green stage, and Z17, T4, and T3 were the characteristic volatiles at the breaking stage, and Z29, Z7, Z22, and C2 were the marker aroma compounds of mature fruit. In HDL1 pepper fruit, Z11 and Z13 were the marker aroma compounds at the green stage, Z29 and Z38 (4-methyl-pentyl 8-methylnon-6-enoate) were the marker volatiles at the breaking stage, and Z17 and T3 were significantly expressed at the mature stage. At the green stage of HDL2, the main marker volatiles were T4 (Himachal-2,4-diene) and Z11, while Z17 was significantly expressed at the breaking stage. Z32 (2-methyl butyl 8-methyl-6-enoate) is a marker aroma compound in mature pepper fruit.

### 2.4. Transcriptome Analysis of Pepper Fruits from Different Samples

To further explore the expression changes of genes related to VACs in pepper fruit, we sequenced the transcriptome of 36 samples. Sequencing quality and data statistics are shown in Appendix A, and pepper reference genome mapping data statistics are shown in Appendix A. We then analyzed the gene expression patterns in different samples of pepper fruits based on the FPKM values of genes (Appendix A). Cluster analysis (Figure 4a) showed that the biological repetition of pepper fruit samples from different samples had good classification. According to the PCA analysis results (PC1: 60.86% and PC2: 25.4%), the samples differed significantly in different varieties and stages (Figure 4b). Then, we used *p* < 0.05 and |log2FC| ≥ 1 as the screening conditions to screen the DEGs between different comparison groups. As shown in (Figure 4c,d), the number of DEGs in the four pepper varieties showed an increasing trend first and then decreased (Figure 4c), indicating that the gene expression change is the most active at the breaking stage, which may be the critical period for pepper to synthesize and metabolize VACs. The comparison of different varieties (Figure 4d) showed that there were more DEGs at the breaking stage and at the mature stage, which might be the critical period for the formation of differential VACs in different varieties. Furthermore, DEGs in different comparison groups showed that the number of down-regulated genes was much higher than that of up-regulated genes.

To further study the change of genes at the transcriptional level, we performed Venn analysis on the DEGs (Figure 5a–c and Appendix A–C). There were 5282, 4790, and 371 common DEGs (Figure 5a–c) at three stage comparison groups of the four varieties (a vs. b, a vs. c, and b vs. c), respectively. While 3273, 4807, and 4548 common DEGs (Appendix A) were in four variety comparison groups at three stages (GJ vs. CTJ, HDL1 vs. CTJ, and HDL2 vs. CTJ), respectively. Furthermore, the results of GO enrichment analysis showed that at level 2, a total of 45 functional subclasses were enriched between the comparison groups at stages (Figure 5d–f), including 22 biological processes, 13 cellular components, and 10 molecular functions. A total of 46 functional subclasses, including 21 biological processes, 13 cellular components, and 12 molecular functions, were enriched among different varieties (Appendix A). KEGG enrichment analysis showed that 117, 114, and 57 metabolic pathways were enriched in three comparison groups at three stages. The pathways related to the synthesis and metabolism of VACs mainly included carotenoid biosynthesis (cann00906), fatty acid elongation (cann00062), alpha-Linolenic acid metabolism (cann00592), phenylpropanoid biosynthesis (cann00940), and valine, leucine and isoleucine degradation (cann00280), fatty acid degradation (cann00071), and fatty acid biosynthesis (cann00061) (Figure 5g–i). These metabolic pathways were significantly enriched (*p* < 0.05) in the comparison group of green and maturity stage, green stage and breaking stage, but were generally not significantly enriched in the comparison group of the maturity and breaking stages, indicating that the gene expression related to the synthesis of VACs mainly occurred in the period before the breaking stage, among which the breaking stage is the critical turning point. Common DEGs among different varieties were enriched to 34 metabolic pathways at the green stage, but most pathways were not significant (Appendix A). At the breaking stage, 116 metabolic pathways (Appendix A) were enriched, among which the pathways related to the synthesis of VACs were significantly enriched, including carotenoid biosynthesis, phenylalanine metabolism (cann00360), and fatty acid degradation. The maturation stage of different pepper variety comparison groups enriched to 115 metabolic pathways (Appendix A). Phenylpropanoid biosynthesis is the most significant enriched metabolic pathway, followed by ubiquinone and other terpenoid-quinone biosynthesis (cann00130) and phenylalanine metabolism, and the volatile ester synthesis of linoleic acid metabolism (cann00591) pathway is significantly enriched.

### 2.5. The Expression Changes of Key Genes in the Biosynthesis Pathway of VACs

Volatile esters (especially branched-chain esters) and aldehydes, which are the most abundant and changing substances in pepper fruits, are mainly produced by the fatty acid pathway (FAP) and branched-chain amino acid pathway (BCAAP). According to transcriptome annotation and homologous sequence alignment, 90 genes involved in FAP and 35 genes involved in BCAAP were identified with differential expression (Figure 6 and Appendix A). In the FAP (Figure 6 (left)), we identified 13 genes encoding LOX, 1 HPL, 11 ADH, and 3 AAT, among which most of the DEGs were significantly downregulated in pepper fruit development (*p* < 0.05), which is consistent with the change of VACs of straight-chain fatty acids such as hexanal and (e)-2-hexenal, indicating that they may play an important role in this process. Similarly, the genes encoding ACC, MAT, FAD, AOS, and ACX showed similar expression patterns. In addition, we found the *LOC107850471* gene encoding SAD, the *LOC107839570* gene encoding 9S-LOX, and the *LOC107867711* gene encoding ADH had significantly high expression at the breaking and maturation stages, indicating that these genes may be associated with the significant accumulation of VACs of straight-chain fatty acids such as hexyl hexanoate in middle and late pepper fruits. Compared with different varieties, most of the DEGs showed lower expression levels compared with CTJ in GJ, HDL1, and HDL2 pepper fruits of the same period.

Branched-chain aminotransferase (BCATs) perform the last step of synthesis of L-leucine, L-isoleucine, and L-valine and the first step of catabolism to produce volatile branched-chain esters in the BCAAP (Figure 6 (right) and Appendix A). Three BCAT genes showed significant differences in different samples, among which *LOC107851037* was significantly downregulated during pepper fruit development, while *LOC107867296*, showed higher expression at the breaking and maturation stages, indicating that the gene of this family may have some control in the generation of volatile esters in BCAAP comprehensively and coordinately. Interestingly, the expression level of *LOC107867296* was significantly higher in HDL2, which may be related to the significant accumulation of 4-methylpentyl 8-methylnonanoate (Z39) in pepper fruit at the breaking stage of HDL2, but was almost undetectable in the other three pepper varieties. As the key enzyme that catalyzes the straight-chain aldehydes to produce acids in BCAAP, seven genes encoding Aldehyde dehydrogenase (ALDH) were identified, and most of these genes’ expression are significantly up-regulated in the fruit development, among which LOC107862407 has more significant expression, indicating that it may have a more critical role in the synthesis of branched ester. In our study, 10 Carboxylesterase (CXE) genes were identified, among which *LOC107872419* and *LOC107859022* gene expression were significantly reduced in the middle and late stages of pepper fruit development, which may be related to the significant increase of most branched-chain esters in pepper fruit at breaking and maturation stages. In addition, we found that the *LOC107867571* and *LOC107859022* genes encoding CXE were more highly expressed in HDL2 than the other three varieties, which may associate with 4-methylpentyl 8-methylnonanoate (Z39) significant accumulation at the mature stage.

Methyl salicylate (Z11, MeSA) is widely found in various fruits and vegetables, often exhibiting green and minty aromas. In this experiment, MeSA mainly accumulated significantly and was the primary characteristic VACs of pepper fruits at the green stage. We detected 12 DEGs involved in MeSA synthesis, including 3 genes encoding DAHP synthase, 3 genes encoding AroE, 1 gene encoding MtSK, 1 gene encoding ICS and 4 genes encoding SAMT, most of these genes had higher expression levels (Appendix A and Appendix A) at green stages. Moreover, comparing CTJ with the other three pepper varieties, we found that one gene encoding DAHP synthase (*LOC107867463*) and two genes encoding SAMT (*LOC107864268* and *LOC107849647*), were significantly more expressed at breaking and mature stages, which was consistent with the accumulation pattern of MeSA, indicating that these genes have a significant regulatory role in the MeSA biosynthesis in pepper fruit.

A total of 20 volatile terpene compounds were detected in this experiment, mainly consisting of monoterpenes and sesquiterpenes. As shown in Figure 7 and Appendix A, our transcriptome data identified 49 differentially expressed genes involved in MVA and MEP pathways. 1-Deoxy-D-xylulose 5-phosphate synthase (DXS), 3-hydroxy-3-methyl-glutaryl-CoA synthase (HMGS), and 3-hydroxy-3-methylglutaryl-CoA reductase (HMGCR) are considered to be the critical rate-limiting enzymes in the MVA and MEP pathways. In this study, we identified three differentially expressed DXS genes, three HMGs, and five HMGCRs. Although the number of these gene families was small, their expression levels have changed significantly in pepper fruits of different samples, which may play an important role in accumulating terpene compounds in pepper fruits. As the last key enzyme for the synthesis of terpenes, 16 TPSs were identified, most of which were significantly expressed at the green stage, and the expression of *LOC107868331*, *LOC107846955*, *LOC107877838*, and *LOC107840471* was more prominent, indicating that they play a more critical role in the synthesis of terpene aromatic compounds. The expression level of *LOC107878220* (TPS) was significantly higher in the fruits of GJ at the breaking and maturation stages than in the other three varieties, which was consistent with the accumulation pattern of T4 (himachali-2,4-diene) in the four pepper varieties. In this experiment, five genes encoding carotenoid cleavage dioxygenase (CCD) were identified, which may be involved in regulating the formation of α-ionone in pepper fruit.

### 2.6. Gene Expression Verified by Real-Time Quantitative PCR (qRT-PCR)

Fifteen key DEGs were selected for qRT-PCR verification analysis, among which SAD, ADH, SAD, HPL, AOS, AOC, FAD, LOX, ACX, BCAT, PD, and FAD in FAP and BCAAP, and TPS, and DXS genes were related to volatile terpenoid synthesis. As shown in Appendix A, the expression profiles of 15 genes were consistent with the expression results of RNA-Seq, indicating that the transcriptome data in this experiment were accurate and reliable.

### 2.7. Correlation Analysis of TFs and Key DEGs in the Synthesis Pathway of Aroma Components

TFs play an important role in the synthesis of plant VACs. A total of 2755 genes were identified as TFs, of which 1965 genes showed significant differences between at least two samples and were further divided into 58 families. Figure 8a shows the cluster heatmap of FPKM values of all DEGs in the same TF family in each group. The results showed that these TF families had significant differential expression in different pepper fruit samples. Furthermore, we analyzed the correlation between the key DEGs in the FAP, BCAAP, and terpenoid biosynthesis pathways and the differential TFs. As shown in Figure 8b, a large number of key DEGs involved in the synthesis of aroma components have a significant correlation with TFs (|r| > 0.9), *p*-value < 0.05), indicating that these transcription factors may be involved in the volatile biosynthesis by regulating the expression of key genes in the VACs synthesis pathway.

### 2.8. Co-Expression Network Module Analysis Identifies DEGs Modules Involved in the Biosynthesis of VACs

To gain insight into the molecular mechanisms regulating the formation of VACs in pepper fruits, we constructed gene networks and co-expression analysis using non-redundant DEGs. The results showed that these DEGs were clustered into ten major modules, and genes in the same module showed higher co-expression and correlation relationships. The correlation analysis between the gene module and trait module with 70 differential VACs as phenotypic traits showed that (Figure 9a) the dark-olive-green module was positively correlated with the accumulation of most aroma components, while the light-yellow module was negatively correlated with the most VACs. The genes in these two modules were used for further cluster analysis, and the results showed that the genes in the dark-olive-green module (Figure 9b) had lower expression levels in CTJ than in other samples, while the genes in the light-yellow module showed the opposite expression patterns, higher expression of genes was mainly in CTJ (Figure 9c).

In order to screen for hub genes associated with the synthesis of VACs, we investigated key genes in the volatile biosynthesis pathway in the dark-olive-green and light-yellow modules and TFs with significant correlation with these essential genes (r > 0.8 or <−0.8, *p*-value < 0.05). A total of 8 TF genes and 3 VACs synthesis-related genes were identified in the dark-olive-green module, and 22 TF genes and 4 VACs synthesis-related genes were identified in the light-yellow module (Appendix A). We then selected these TFs and aroma synthesis-related genes as essential hub genes to construct a co-expression network of the dark-olive-green and light-yellow modules. The results showed that (Figure 9d,e) most of these essential hub genes are closely related to other genes in the plate and are closely linked. It can be speculated that they play an essential role in regulating the synthesis of aroma substances. Meanwhile, many TFs were identified as crucial hub genes, indicating that they play an essential role in forming VACs in pepper fruits. Notably, the genes directly involved in the aroma volatile synthesis pathway did not largely cluster in the dark-olive-green and light-yellow modules, possibly due to the low expression level of most genes involved in the volatile synthesis and the diversity of volatile aroma products.

## 3. Discussion

### 3.1. Aroma Perception from Dynamic Changes of VACs in Pepper Fruits

The volatile components and relative content are important reasons for people’s unique aroma perception of different types of fruits and vegetables [29]. In this study, we detected more than 140 VACs with different contents in a variety of pepper fruits, indicating that the aroma differences were mainly affected by the relative content of VACs. We found that HDL1 (Hainan Huangdenglong pepper) and HDL2 (hybrid Huangdenglong pepper) are both processed varieties of chili sauce, and their genetic relationship is similar. However, HDL1 mature fruit has a higher content of flavor components, and the accumulation pattern of HDL2 flavor components is the opposite, which may be the reason why its chili sauce is more popular with consumers. Studies have shown that straight and branched-chain esters can give *Capsicum chinense* pepper fruits a solid fruity/exotic aroma [23,30]. In our study, esters were the most critical aroma components in four pepper varieties, similar to the results of previous studies on Cachucha peppers [23], Habanero pepper [31], and Tabasco pepper [32] and accumulated in the late stage of most fruit development in HDL1 and GJ (ghost pepper), which is consistent with the accumulation pattern of fruity esters [33,34], further explaining the cause of the processing aroma of HDL1 mature fruit. However, the accumulation of esters gradually decreased during HDL2 and increased first and then decreased during CTJ (Chaotian pepper) fruit development, indicating the accumulation pattern difference of esters in different cultivars and developmental stages. The content of esters in HDL 1 mature fruit was significantly higher than other varieties at the same stage, which may be an important reason for for its rich aroma. Aldehydes were the main aroma components at the green stage of pepper fruits and showed a decreasing expression trend with fruit development, which may be related to the gradual transformation of aldehydes into esters with fruit ripening. Terpenes, showing floral and fruity aroma in tea, were detected in pepper fruit in this experiment, and the accumulation pattern was affected by varieties and development stages as well as esters. Although their concentration is usually low, these compounds usually have a low aroma threshold [35], which has an important impact on the aroma quality of pepper.

### 3.2. Analysis of Marker Aroma Compounds in Pepper Fruits from Different Samples

Generally, the special flavor of vegetables, fruits, and tea comes from the marker aroma compounds. (E)-2-hexenal and (Z)-3-hexenal are the most important aromatic compounds with strong green grass and green leaf odor in cherry tomato [36], while linalool oxide and linalool are the main components of citrus flower fragrance [37], and hexyl 2-methylbutyrate is the main reason for the “honey crisp” apple flavor [34]. The characteristic components (E, E)-2,4-hexagon make Jingshan green tea have a floral and fatty flavor [38]. In Brazil’s tabasco pepper, α-Pinene, hexyl 2-methylbutyrate, hexyl 3-methylbutanoate, and hexyl butanoate are the main contributors to the typical sweet, herbal, and pepper-like flavors [32]. Our research confirmed methyl salicylate (Z11) was the main marker aroma compound at the green stage of four pepper varieties and described as green and sweet in the Brazilian tabasco pepper [32], hexanal (Q3) described in cherry tomatoes as having the smell of freshly cut grass [36], are another characteristic VACs of CTJ and GJ at green stage, however 4-methylpentyl-2-methylbutyrate (Z12) not only accumulated significantly at the green phase of GJ, and was the characteristic volatile of CTJ mature fruit with herbal flavor [32]. Moreover, as one of the main marker volatiles of CTJ and HDL1, 4-methylpentyl 3-methylbutanoate (Z13) may show the same odor as Z12. The study found that the characteristic component trans-2-hexenylisovalerate (Z17) with green and fruit-like odor accumulated significantly at the green stage of HDL2 and breaking stage of GJ, while the hexyl hexanoate (Z27) with an apple fruit flavor is the main marker volatile component in the mature fruit of HDL1, and is usually described as fresh cut grass and vegetable flavor [34]. The above showed that the marker aroma components are different in pepper fruits of different varieties and development stages, which can be used as important indicators to study the flavor quality of pepper. However, how they affect the specific aroma quality of these pepper fruits still needs to be determined. Further sensory evaluation and gas-chromatography–olfactometry (GC-O) methods are needed to study the specific effects of these volatile aroma substances on the aroma of pepper fruit [33,39].

### 3.3. Many DEGs and TFs Are Involved in the Synthesis of VACs in Pepper Fruits

The expression of key genes in amino acids, fatty acids, MVA, and MEP metabolism are an essential factor affecting the formation of aldehydes, alcohols, and esters producing VACs [9,14]. The Lipoxygenase pathway is the key pathway for the biosynthesis of straight-chain aldehydes, alcohols, and esters, including four-step enzymatic reactions, in which LOX, as the first step in the synthesis of straight-chain volatile fatty acid derivatives, has been proven to be able to catalyze the oxidation of polyunsaturated fatty acids in apples to produce volatile C6 aldehydes [40]. HPL catalyzes the conversion of LOX reaction products into aldehydes, and its silencing significantly reduces the production of C6 volatiles in olive [41], such as hexanol, hexanal, and (E)-2-hexanol. ADH catalyzes the reduction of straight-chain aldehydes in fruits to corresponding straight-chain alcohols, which has been proven to be closely related to the formation of watermelon C9 alcohols [42]. AAT catalyzes linear alcohols and acyl carrier proteins (ACPs) to form volatile straight-chain esters in many horticultural plants, such as flowers, fruits, and vegetables [6]. Our study identified 13 LOX, 1 HPL, 11 ADH, and 3 AAT genes with different expression patterns in pepper fruits of different varieties and developmental stages with similar trends to some volatile compounds, such as hexanal, (e)-2-hexenal, hexanoic acid-hexyl ester. This suggests that these DEGs may regulate the production of straight-chain volatiles in pepper fruit.

Branched-chain esters are the most critical VACs in pepper fruits in this study, mainly produced by BCAAP. As the last enzyme in the BCAAP, the catalytic product branched-chain amino acid of BCAT provides a precursor for the synthesis of volatile branched-chain compounds [15]. Studies have shown that BCAT is related to the formation of branched-chain esters in fruits such as bananas [43], melons [16], and apples [29]. Another study on peppers showed that the accumulation of most branched esters was significantly positively correlated with the *CcBcat4* gene [33]. Our study confirmed three differentially expressed BCAT, of which the expression pattern of *LOC107867296* was consistent with the changing trend of 4-methylpentyl-8-methyl-nonanoate (Z39) content, indicating that it may play a key regulatory role in Z39. CXE is a key enzyme for the synthesis of branched-chain esters, which has been shown to hydrolyze flavor esters in peach [17], strawberry [44], and tomato [45]. In this study, 10 genes encoding CXE were identified, of which *LOC107867571* and *LOC107859022* encoding CXE gene decreased significantly in the breaking and maturation stages of pepper fruit, which may be related to the accumulation of volatile branched-chain esters in pepper fruit in the middle and late growth. Methyl salicylate (Z11) produced by SAMT-catalyzed SA is the most important aromatic compound in pepper fruit [46]. We identified four SAMT-coding genes; however, their expression pattern is not completely consistent with the accumulation pattern of methyl salicylate (MeSA), which indicates that the synthesis of MeSA in pepper fruit may be regulated by other factors that need further study. As the most important enzyme-producing terpene compounds and the gene family with diverse functions in plants [14], TPS directly participated in the synthesis of monoterpene, sesquiterpene, and carotenoid derivatives, the main VACs of HDL1 and GJ fruits. In our experiment, 16 genes encoding TPS were identified as having different expression patterns, which may be related to their functions and the diversity of substrates produced. In addition, we found many other key DEGs related to the synthesis pathway of volatile aromatic compounds, which may play an important role in regulating the synthesis of volatile aromatic compounds in pepper fruits.

TFs play an essential role in the metabolism of aroma volatiles [47]. It has been reported that *MYB* regulates the production of volatile compounds in strawberries [48], *PAP1* can enhance the synthesis of phenylpropanoids and terpenoids in roses [49], the synthesis of volatile esters and monoterpenoids in kiwifruit is closely related to *NAC* transcription factors [50], *AP2/ERF* transcription factors have been proven related to the production of volatile terpenes in sweet orange fruits [20], and in tomatoes, *RIN* transcription factors can regulate the production of volatile aldehydes and alcohols in fruits by regulating the expression of key genes in the LOX pathway [51]. In our experiment, we found that many TFs (*C2H2*, *C3H*, and *MYB*, etc.) were significantly correlated with the critical DEGs and differential metabolites involved in the synthesis of aroma components, indicating that these TFs may be involved in the synthesis of aroma components by regulating the expression of critical genes in the synthesis pathway of VACs in pepper fruits. Therefore, future studies will focus on the specific functions of these TFs in regulating the synthesis of volatile aroma compounds in pepper.

## 4. Material and Methods

### 4.1. Plant Materials

The four pepper varieties used in this experiment were preserved by the Pepper Research Group of Hainan University respectively, including “HDL1” (*Capsicum chinense* L., local varieties unique to Hainan Province, China), “HDL2” (*Capsicum chinense* L., an exotic hybrid whose fruit shape is different from HDL1 is being popularized in Hainan Province), “GJ” (*Capsicum chinense* L., a highly hot variety originating from India), “CTJ” (*Capsicum annuum* L., an annual dried pepper variety), specific sample phenotype information is shown in Appendix A and Figure 1. The four pepper varieties are representative in terms of their close affinity, economic species, and varieties, which can reflect the cultivation of pepper in China. All peppers were planted in plant growth rooms in October 2021 (temperature: 23 ± 0.5 °C; light duration: 16 h; light intensity: 6500lx), with ten plants for each cultivar, and cultivation and management measures were consistent. From December 2021 to February 2022, fruits of four pepper varieties were collected at the green, breaking, and maturation stages based on fruit size and color. Thirty fruits were randomly selected from each stage and divided into three biological replicates, totaling 36 fruit samples. Each fruit sample was homogenized by a mixing method, quickly frozen in liquid nitrogen, and stored in an ultra-low temperature refrigerator at −80 °C for subsequent transcriptome and metabolomics analysis.

### 4.2. Untargeted Metabolome Analysis

#### 4.2.1. Sample Preparation and Extraction

Shanghai OE Biotech Co., Ltd. completed untargeted metabolome sequencing. HS-SPME extracted VACs from pepper fruit samples: Fruit samples are frozen fresh fruit and were immediately powdered with a grinder (MM400, Retsch, Dusseldorf, GER), then defrosted at room temperature prior to analysis. Accurately weigh the 1 g of sample powder and add to an ethanol solution containing 20 μg/mL n-alkanes (C7–C40) as an internal standard (configuration method: 980 μL ethanol solution (99.7%) + 20 μL n-alkanes mother liquor (1 mg/mL)) and then transfer to a sealed headspace injection bottle (20 mL, Agilent, Santa Clara, CA, USA) for the release of volatiles. The headspace bottle was equilibrated at a temperature of 60 °C and a shaking speed of 450 rpm for 10 min. Then the extraction head (50/30 μm DVB/CAR/PDMS, Sigma, Shanghai, China) was inserted into the headspace part of the sample and extracted for 60 min [52,53].

#### 4.2.2. GC-MS Analysis of the VACs

Agilent 7890B chromatography and a 5977B mass spectrometer equipped with a DB-5MS capillary column (30 m × 0.25 mm × 0.25 μm, Agilent J&W Scientific, Folsom, CA, USA) carried out the qualitative and quantitative analysis of volatile aromatic compounds. The extracted sample was directly injected into the injection port of the gas chromatograph–mass spectrometer, desorbed at 250 °C for 5 min, and the injection port temperature was set at 230 °C. The high-purity helium (Purity ≥ 99.999%) was used as the carrier gas. The initial oven temperature was set at 40 °C for 1 min, increased to 230 °C at 4 °C/min and held for 1.5 min, then increased to 250 °C at a rate of 10 °C/min and kept for 2 min. Mass spectrometry was recorded in the ionization mode of electron impact ion source (EI) energy of 70 eV. The ion source temperature was set to 230 °C, and the quadrupole temperature was 150 °C. Mass spectrometry data were extracted by full scan mode (SCAN) with a mass scanning range of 40–500 m/z. In the process of mass spectrometry analysis, all samples were mixed as quality control (QC) samples to test the stability of the system mass spectrometry platform during the whole experiment. MS-DIAL software analyzed the GC-MS raw data for peak detection, peak recognition, MS2 Dec deconvolution, and peak alignment. They then compared with MS in the NIST database to determine the species of VACs. VACs with MS matching scores greater than 80% were retained, and the final substance type was determined after manual identification and comparison. The relative content of volatiles was calculated by the standard internal method. The formula is as follows:$$f\left[\mathrm{relative}\;\mathrm{content}\;\mathrm{of}\;\mathrm{volatiles}\;\left(\mu g/\mathrm{kg}\right)\;\right]=\frac{\mathrm{peak}\;\mathrm{area}\;\mathrm{of}\;\mathrm{target}\times 0.1\;\mu g}{\mathrm{peak}\;\mathrm{area}\;\mathrm{of}\;\mathrm{internal}\;\mathrm{standard}\times 1000\;g}$$

#### 4.2.3. Screening of Differential VACs and Marker Aroma Compounds

Multivariate statistical analysis was used to screen the differential VACs in samples: The relative content of VACs was used as the analysis data, and the PLS-DA model of VACs in different pepper fruit samples was constructed by (PLS-DA). The VACs in the model were screened by variable weight value (VIP) > 1, and a T-test was performed to verify its significance.

Reference: Liu et al.’s method for screening marker aroma compounds [24].

### 4.3. Transcriptome Sequencing

#### 4.3.1. RNA Extraction and Library Preparation

Total RNA was extracted from 36 samples (4 pepper varieties × 3 developmental stages × 3 biological replicates) using FastPure^®^ Plant Total RNA Isolation Kit (Vazyme, China). The purity and quantitative analysis of extracted RNA were evaluated using the NanoDrop 2000 ultraviolet spectrophotometer (Science Corporation, USA). The integrity of RNA was evaluated using the Agilent 2100 Bioanalyzer (Agilent Technologies, Santa Clara, CA, USA). The qualified total RNA was used to construct the cDNA library.

#### 4.3.2. RNA Sequencing Analysis

The transcriptome sequencing and DEGs expression analysis were performed by Shanghai OE Biotech Co., Ltd. (Shanghai, China). Pre-process the raw data to remove low-quality reads containing Ploy-N. The obtained clean reads were mapped to the pepper genome Pepper Zunla_1_Ref_v1.0 (https://www.ncbi.nlm.nih.gov/data-hub/genome/GCF_000710875.1/ (accessed on 30 March 2022).) using the default parameter HISAT2 software. Then the transcript assembly was performed, and the transcript abundance was quantified. The calculation formula of transcript abundance FPKM value is as follows:$$f\left(\mathrm{FPKM}\right)=\frac{\mathrm{cDNA}\;\mathrm{Fragments}}{\mathrm{Mapped}\;\mathrm{Fragments}\;\left(\mathrm{Millions}\right)\times \mathrm{Transcript}\;\mathrm{Length}\;\left(\mathrm{kb}\right)}\;$$

#### 4.3.3. DEGs Analysis

DEGs expression analysis was performed using DESeq (2012) R software and identified as DEGs under the conditions of *p*-value < 0.05 and |log2FoldChange| > 1. GO (Gene Ontology) enrichment analysis was performed using GOseq R package software, and KEGG (Kyoto Encyclopedia of Genes and Genomes) signaling pathway enrichment analysis was performed using KOBAS 2.0 software.

### 4.4. qRT-PCR Validation of Transcriptome Data

Total RNA was extracted from 36 pepper samples using FastPure^®^ Plant Total RNA Isolation Kit (Vazyme, China), and reverse transcription was performed using HiScript^®^ III All-in-one RT SuperMix (Vazyme, China). 15 DEGs related to the synthesis of VACs were selected for qPCR, and gene-specific primers were designed using Primer3 software (Appendix A). The qRT-PCR reaction was performed using Quant Studio 3 real-time fluorescence quantitative PCR instrument (ABI, Shanghai, China) and ChamQ Universal SYBR qPCR Master Mix. Quantitative Expression Data of Genes Calculated by 2^−ΔΔCt^ Method with Actin Reference Gene.

### 4.5. Construction of Co-Expression Analysis by WGCNA

The co-expression network was constructed with 6204 DEGs (*p*-value < 0.05 and |log2FoldChange| > 1 with FPKM ≥ 1 and standard deviation ≤ 0.8), and the weighted gene co-expression network analysis was performed using the R software package. In order to obtain genes related to the synthesis of aroma volatiles, the relative content data of 70 differential metabolites were used as phenotypic traits to correlate with modules, and the significance of DEGs analyzed the correlation between trait data and gene expression data. We used Cytoscape3.9.1 software to visualize the co-expression module.

### 4.6. Statistical Analysis

Trials were conducted entirely randomized, with three replicates per trial and data expressed as mean ± standard deviation (SD). SPSS v19.0 software was used to perform statistical analysis and one-way analysis of variance (ANOVA) of the data, and the Duncan test was used to evaluate the significant difference between samples (*p* < 0.05). Graphics were mainly drawn using Tbtools, Origin 2021, Excel 2021, Cytoscape 3.9.1, prism8, and other drawing software.

## 5. Conclusions

In summary, we found significant quantitative differences in VACs from several Chinese spicy peppers and one India pepper, and volatile esters were the most crucial aroma components in different variety and developmental stage pepper fruits. The higher accumulation of total VACs and volatile esters in Hainan Huangdenglong pepper during fruit development may be the reason for better processing aroma quality. We found that the marker aroma compounds in different pepper fruit samples were different, which may have a crucial effect on the aroma of pepper fruit. Meanwhile, many DEGs and TFs related to volatile compound biosynthesis were identified in pepper fruit. Moreover, the breaking stage may be critical for forming aroma compounds. At last, the comprehensive analysis of transcriptome and metabolomics identified potential candidate genes and transcription factors that may be involved in volatile compound biosynthesis in pepper, which can be essential candidate genes for subsequent functional verification studies. This paper is the first attempt to systematically study the metabolic pathways and molecular regulation mechanisms of VACs in pepper, which provides a theoretical basis for the molecular breeding of high-quality pepper.

## Figures and Tables

**Figure 1 ijms-24-07901-f001:**
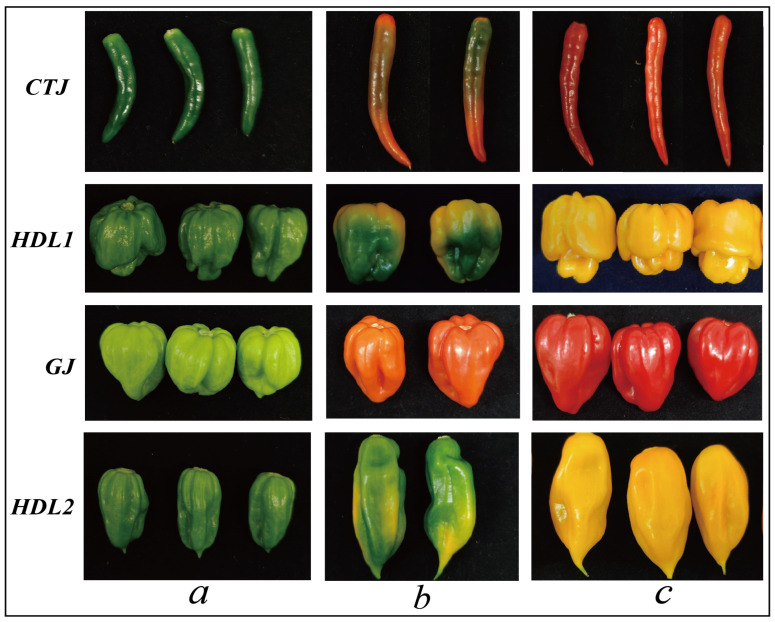
Phenotype plots of pepper fruits from different samples. CTJ: Chaotian pepper; HDL1: Hainan Huangdenglong pepper; GJ: ghost pepper; HDL2: hybrid Huangdenglong pepper; “a”: green stage; “b”: breaking stage; “c”: maturation stage.

**Figure 2 ijms-24-07901-f002:**
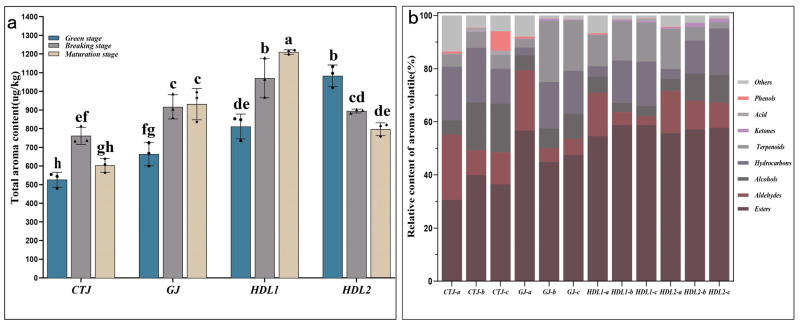
Volatile aroma composition in different samples. (**a**) Total volatile aroma content (μg/kg) in pepper fruit of different samples. Error bars show ±SE from three biological replicates. Significant differences are shown with different letters above the bars (*p* < 0.05); (**b**) the main volatile aroma component categories and their relative content (%) in different pepper fruit samples.

**Figure 3 ijms-24-07901-f003:**
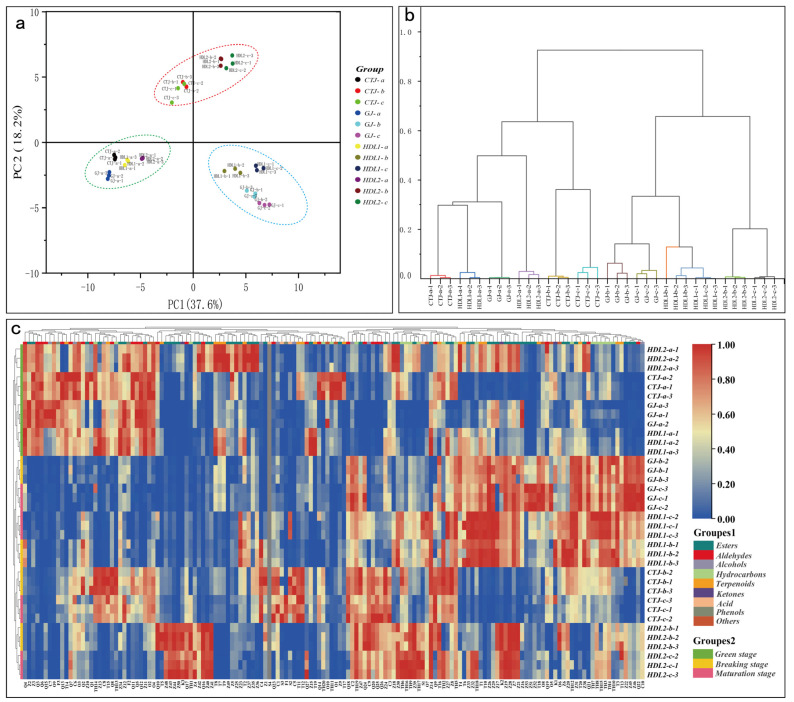
Characterization of VACs from different pepper samples; (**a**) PCA of VACs from different pepper samples; (**b**) HCA of VACs in different samples; (**c**) heat map of the species and relative contents (μg/kg) of volatiles in different samples. (The heatmap was drawn using log2-based relative content values. Red represents high expression, and blue represents low expression).

**Figure 4 ijms-24-07901-f004:**
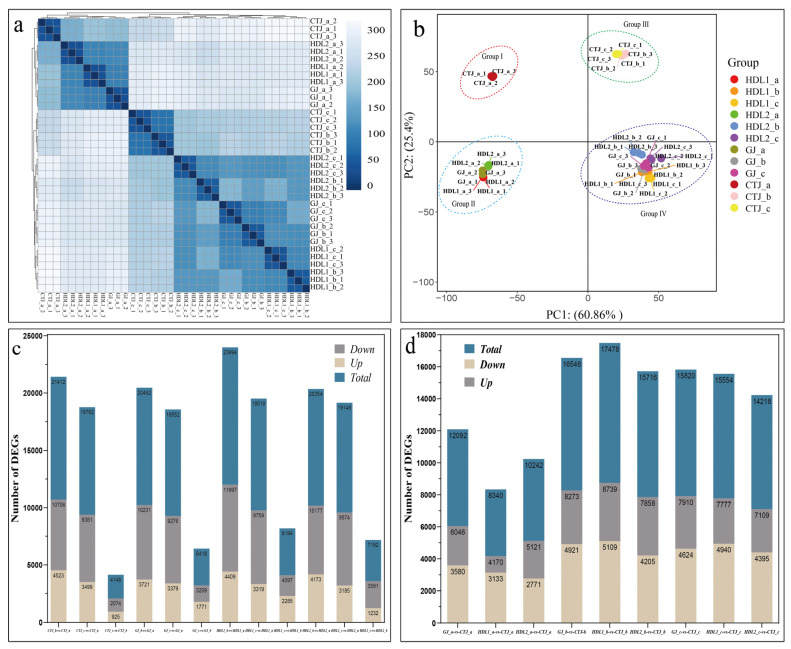
Summary analysis of transcriptomes. (**a**) Cluster plot of the transcriptome sequencing samples; (**b**) PCA of 36 pepper fruit samples; (**c**) the number of DEGs during pepper fruit development; (**d**) the number of DEGs between the four pepper varieties.

**Figure 5 ijms-24-07901-f005:**
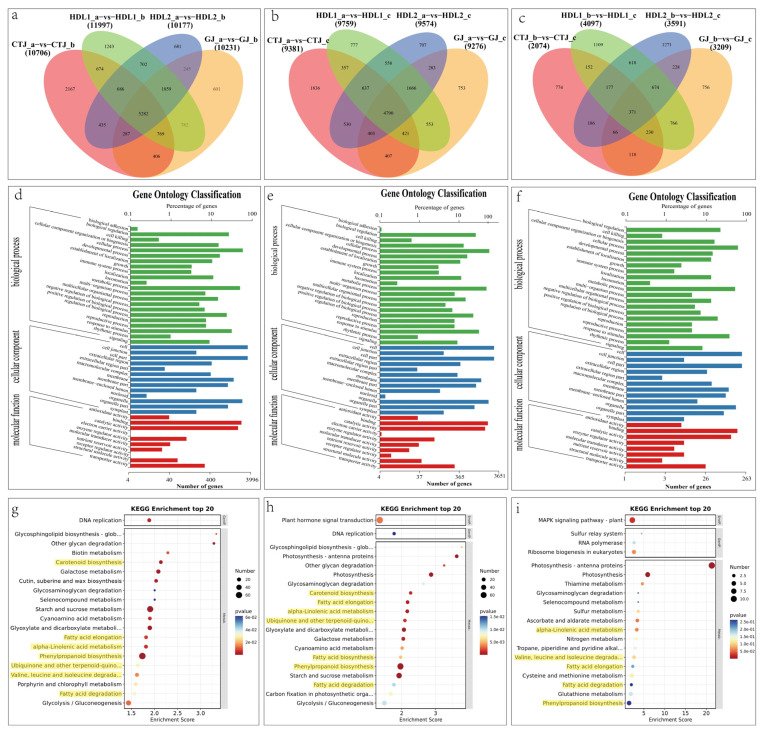
Functional enrichment analysis of DEGs in different developmental stages. (**a**,**b**) Venn diagram of the DEGs four in pepper varieties at different growth periods; (**d**–**f**) gene ontology (GO) classifications of overlapping DEGs in (**a**–**c**); (**g**–**i**) Kyoto Encyclopedia of Genes and Genomes (KEGG) pathway enrichment of DEGs in (**a**–**c**).

**Figure 6 ijms-24-07901-f006:**
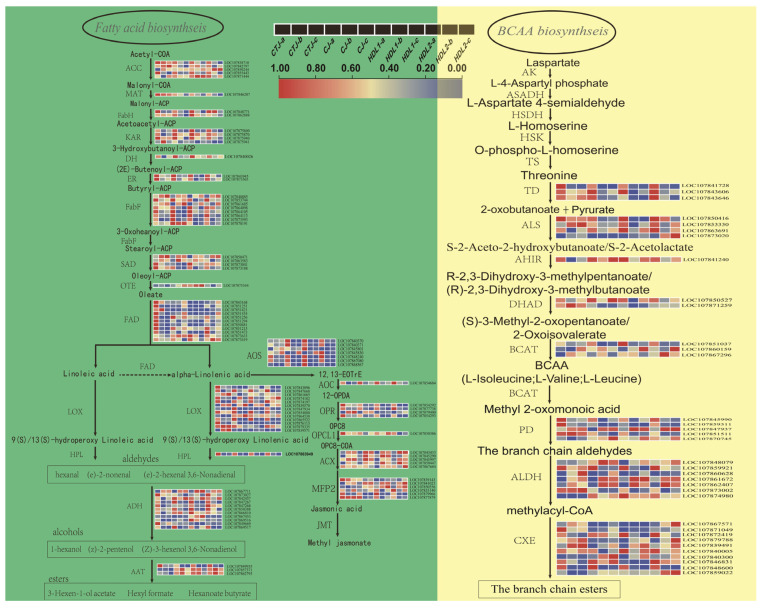
Expression profiles of volatile-related genes involved in the fatty acid (**left**) and branched-chain amino acid (**right**) metabolism pathway in different pepper fruit samples (red represents high expression, and blue represents low expression). ACC: acetyl-CoA carboxylase; MAT: malonyl-CoA ACP transacylase; FabH: 3-ketoacyl ACP synthase II; KAR: ketoacyl ACP reductase; DH: 3-hydroxyacyl ACP dehydratase; ER: 2,3-trans-enoyl ACP reductase; FabF: 3-ketoacyl ACP synthase; SAD: stereate ACP desaturase; OTE: oleate-ACP thioesterase; FAD: oleate and linoleate desaturase; LOX: lipoxygenase; HPL: hydroperoxide lyase; ADH: alcohol dehydrogenase; AAT: alcohol acyl transferase; AOS: allene oxide synthase; AOC: allene oxide cyclase; OPR: 12-oxophytodienoate reductase; OPCL1: OPC-8:0 CoA ligase 1; ACX: acyl-CoA oxidase; MFP2: enoyl-CoA hydratas; JMT: jasmonic acid carboxyl methyltransferase; AK: aspartate kinase; ASADH: aspartate semialdehyde dehydrogenase; HSK: homoserine kinase; TS: threonine synthase; TD: threonine deaminase; ALS: acetolactic synthetase; AHIR: acetohydroacid isomeroreductase; DHAD: dihydroxy acid dehydratase; BCAT: branched-chain aminotransferase; PD: pyruvate decarboxylase; ALDH: aldehyde dehydrogenase; CXE: carboxylesterase.

**Figure 7 ijms-24-07901-f007:**
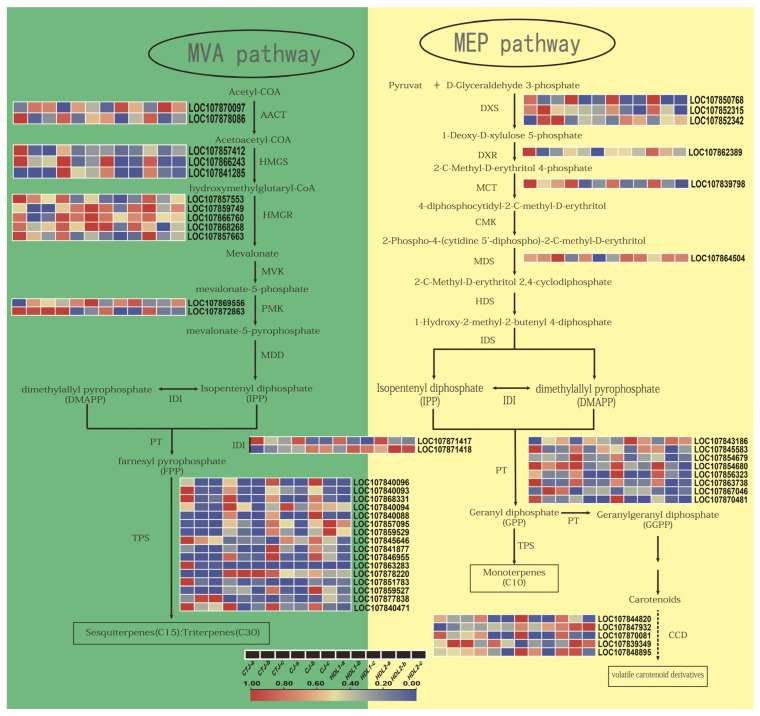
Expression profiles of volatile-related genes involved in the MVA (**left**) and MEP (**right**) metabolic pathways in the different pepper fruit samples (red represents high expression, and blue represents low expression). AACT: acetoacetyl-CoA thiolase; HMGS: 3-hydroxy-3-methyl-glutaryl-CoA synthase; HMGR: 3-hydroxy-3-methylglutaryl-CoA reductase; MVK: mevalonate kinase; PMK: phosphomevalonate kinase; MDD: mevalonate-5-diphosphate decarboxylase; IDI: isopentenyl pyrophosphate isomerase; DXS: 1-deoxy-D-xylulose 5-phosphate synthase; DXR: 1-deoxy-D-xylulose 5-phosphate reductoisomerase; MCT: 2-C-methyl-D-erythritol 4-phosphate cytidylyltransferase; CMK: 4-(cytidine 5′-diphospho)-2-C-methyl-D-erythritol kinase; MDS: 2-C-methyl-D-erythritol 2,4-cyclodiphosphate synthase; HDS: 4-hydroxy-3-methylbut-2-en-1-yl diphosphate synthase; IDS: isopentenyl diphosphate synthase; PT: prenyl transferase; TPS: terpene synthase; CCD: carotenoid cleavage dioxygenase.

**Figure 8 ijms-24-07901-f008:**
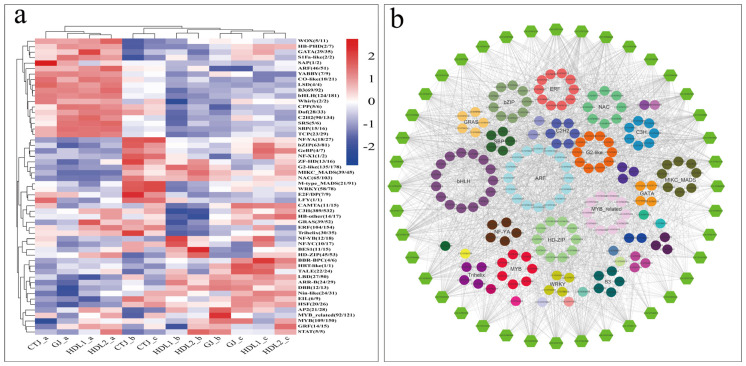
Expression patterns of TFs’ families. (**a**) Heatmaps of different TF families in different samples (the color represents the total FPKM of all differential TFs of a particular TF family. The numbers in brackets indicate the number of differentially expressed members and the total number of the TF family identified in this study). (**b**) Transcriptional regulation of TFs associated with the biosynthesis of volatile aroma compounds (hexagons represent structural genes associated with the fragrance substance biosynthesis pathway, and spheres represent different transcription factors. Colored lines represent the expression correlation between TFs and genes involved in aroma compound biosynthesis (Pearson’s correlation test, *p* < 0.05, correlation coefficient (|r| > 0.9)).

**Figure 9 ijms-24-07901-f009:**
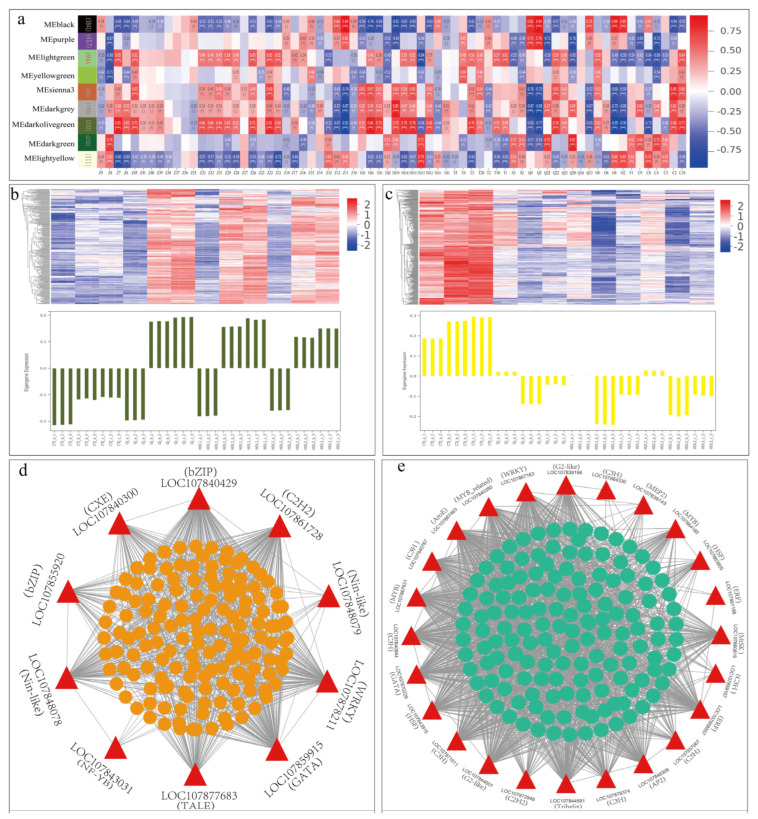
Co-expression module identification based on weighted gene co-expression network analysis (WGCNA). (**a**) Module-trait associations based on Pearson correlations. Each row represents a module, and the number of genes in each module is shown on the left. Each column represents a volatile-based aroma compound (“*” represents the significance between groups, the more the number, the more significant). (**b**,**c**) Heatmaps of genes in modules Dark-olive-green (**b**) and light-yellow. (**c**) Red represents high expression and blue represents low expression. (**d**) Gene co-expression network of the Dark-olive-green module. Co-expression networks were constructed using the top 150 genes based on module membership (kME) values. (**e**) Gene co-expression net-work of the light-yellow module (The red triangles represent the key hub genes in the module).

**Table 1 ijms-24-07901-t001:** A total of 149 volatile aroma compounds were detected in pepper fruits.

RT	CAS	Stage		Green Stage				Breaking Stage				Maturation Stage			
		Aroma Volatile Profiles (μg/Kg)		CTJ	GJ	HDL-1	HDL-2	CTJ	GJ	HDL-1	HDL-2	CTJ	GJ	HDL-1	HDL-2
		Esters	Code												
7.209	629-33-4	formic acid, hexyl ester	Z1	-	-	-	-	0.83 ± 0.15	-	0.3 ± 0.03	0.57 ± 0.15	0.29 ± 0.12	-	0.1 ± 0.01	0.19 ± 0.01
7.734	599-04-2	pantolactone	Z2	2.18 ± 0.53	7.69 ± 0.24	3.27 ± 0.04	3.59 ± 0.1	0.17 ± 0.07	-	0.09 ± 0.04	0.02 ± 0	0.05 ± 0.01	0.14 ± 0.11	0.02 ± 0	-
8.737	106-70-7	hexanoic acid, methyl ester	Z3	5.21 ± 0.82	3.17 ± 0.19	3.79 ± 0.12	2.99 ± 0.34	0.14 ± 0.03	0.03 ± 0	0.75 ± 0.07	0.3 ± 0.08	0.1 ± 0.04	0.02 ± 0.01	0.18 ± 0.04	0.11 ± 0.03
12.919	13574-81-7	n-cbz-glycylglycine p-nitrophenyl ester	Z4	1.73 ± 0.36	0.86 ± 0.18	0.05 ± 0.03	1.1 ± 0.09	0.68 ± 0.27	-	-	2.05 ± 0.04	0.42 ± 0.03	-	-	4.53 ± 1.53
16.01	111-11-5	octanoic acid, methyl ester	Z5	0.62 ± 0.15	0.22 ± 0.04	0.17 ± 0.02	0.45 ± 0.02	0.09 ± 0.01	0.03 ± 0.01	0.15 ± 0.01	0.16 ± 0.05	0.09 ± 0.05	0.17 ± 0.01	0.08 ± 0.02	0.15 ± 0.03
16.568	41519-23-7	cis-3-hexenyl iso-butyrate	Z6	0.55 ± 0.03	0.79 ± 0.08	1.1 ± 0.06	2.97 ± 0.16	3 ± 0.16	3.76 ± 0.13	8.98 ± 0.2	4.48 ± 0.4	1.02 ± 0.24	2.06 ± 0.35	5.32 ± 0.54	2.14 ± 0.2
16.78	25415-62-7	n-amyl isovalerate	Z7	0.21 ± 0.02	12.51 ± 1.61	2.67 ± 0.25	28.52 ± 1	1.72 ± 0.02	35.59 ± 26.48	55.64 ± 2.57	67.49 ± 2.47	0.76 ± 0.15	56.77 ± 10.06	49.68 ± 1.9	55.65 ± 3.55
17.219	35852-44-9	4-Methylpentyl isobutyrate	Z8	20.54 ± 1.68	0.22 ± 0.04	3.36 ± 0.23	1.19 ± 0.09	31.63 ± 0.17	8.66 ± 0.37	12.35 ± 0.48	4.96 ± 0.43	14.57 ± 4.46	5.15 ± 0.96	16.59 ± 0.75	3.43 ± 0.37
18.272	53398-84-8	butanoic acid, 3-hexenyl ester, (e)-	Z9	6.92 ± 0.72	0.16 ± 0.01	0.56 ± 0.06	0.41 ± 0.04	1.05 ± 0.03	0.44 ± 0.05	0.83 ± 0.06	0.62 ± 0.02	0.29 ± 0.09	0.13 ± 0.02	0.5 ± 0.03	0.16 ± 0.01
18.5	2639-63-6	butanoic acid, hexyl ester	Z10	9.2 ± 0.69	0.05 ± 0.01	3.19 ± 0.19	1.13 ± 0.11	1.68 ± 0.02	0.14 ± 0.04	3.06 ± 0.04	0.74 ± 0.52	0.86 ± 0.25	0.08 ± 0.03	1.67 ± 0.53	1.12 ± 0.18
18.568	119-36-8	methyl salicylate	Z11	72.54 ± 5.96	103.55 ± 8.52	119.37 ± 10.73	83.51 ± 6.14	52.21 ± 5.11	0.03 ± 0.02	0.02 ± 0	3.32 ± 0.71	27.63 ± 1.56	0.07 ± 0.05	-	-
18.79	35852-40-5	4-methylpentyl 2-methylbutanoate	Z12	10.23 ± 0.77	91.76 ± 8.94	50.82 ± 5.06	42.88 ± 25.07	8.31 ± 5.71	5.74 ± 2.19	2.51 ± 0.81	4.88 ± 1.37	71.22 ± 10.1	4.57 ± 3.6	1.51 ± 0.36	2.8 ± 1.16
19.041	850309-45-4	4-methylpentyl 3-methylbutanoate	Z13	10.58 ± 0.84	-	95.62 ± 8.49	1.04 ± 0.88	77.66 ± 0.19	0.07 ± 0	0.03 ± 0.01	0.04 ± 0.02	44.76 ± 6.42	0.06 ± 0.02	0.02 ± 0.01	0.08 ± 0
19.86	53398-85-9	cis-3-hexenyl-.alpha.-methylbutyrate	Z14	2.44 ± 0.23	7.05 ± 0.74	19.5 ± 2.43	0.25 ± 0.01	18.21 ± 0.27	0.01 ± 0	17.49 ± 0.73	19.9 ± 0.42	7.64 ± 1.26	-	15.62 ± 0.49	19.34 ± 0.35
20.05	10032-15-2	butanoic acid, 2-methyl-, hexyl ester	Z15	1.82 ± 0.14	2.58 ± 0.55	41.66 ± 4.86	0.15 ± 0.07	7.39 ± 0.08	5.12 ± 0.13	-	0.05 ± 0.03	4.38 ± 0.82	4.17 ± 0.32	-	0.03 ± 0
20.374	10032-13-0	butanoic acid, 3-methyl-, hexyl ester	Z16	0.06 ± 0.01	44.37 ± 5.86	1.65 ± 0.15	0.86 ± 0.04	0.13 ± 0.02	18.23 ± 2.74	7.1 ± 1.13	4.76 ± 3.09	-	15.05 ± 1.06	6.36 ± 0.32	0.77 ± 0.67
20.486	68698-59-9	trans-2-hexenyl isovalerate	Z17	0.1 ± 0.01	43.22 ± 7.77	-	122.59 ± 26.66	0.19 ± 0	52.85 ± 0.19	111.11 ± 56.38	112.48 ± 2.39	0.11 ± 0.02	0.06 ± 0.01	146.56 ± 10.19	26.76 ± 3.09
21.241	1191-02-2	4-decenoic acid, methyl ester	Z18	2.49 ± 0.16	1.72 ± 0.08	2.7 ± 0.35	9.81 ± 0.51	0.43 ± 0.03	0.02 ± 0	4.99 ± 0.17	4.25 ± 1.38	0.15 ± 0.05	0.02 ± 0.01	2.07 ± 0.38	2.94 ± 0.57
21.804		3-methylbut-2-enoic acid, 2-methyl-pentyl ester	Z19	0.2 ± 0.01	0.51 ± 0.1	0.45 ± 0.08	4.37 ± 0.16	0.03 ± 0	9.7 ± 0.46	5.71 ± 0.35	8.73 ± 0.17	0.01 ± 0	10.84 ± 0.64	6.35 ± 0.15	12.17 ± 0.42
21.902	1117-59-5	hexyl n-valerate	Z20	0.37 ± 0.04	-	0.39 ± 0.07	0.27 ± 0.03	1.55 ± 0.03	0.48 ± 0.06	2.2 ± 0.28	1.64 ± 0.04	0.62 ± 0.13	0.34 ± 0.02	1.63 ± 0.1	0.98 ± 0.09
22.001	1215128-03-2	6-methylhept-4-en-1-yl isobutyrate	Z21	0.02 ± 0	1.81 ± 0.44	1.57 ± 0.28	13.3 ± 0.46	0.07 ± 0.01	6.92 ± 5.71	9.19 ± 0.59	17.52 ± 0.37	0.04 ± 0.01	11.69 ± 0.22	9.82 ± 0.41	14.25 ± 0.88
22.094		amyl tiglate, 4-methyl-	Z22	-	-	-	-	-	59.59 ± 2.08	23.78 ± 1.21	11.05 ± 0.12	0.01 ± 0	70.3 ± 2.84	32.81 ± 1.25	20.99 ± 0.47
22.648	1215127-79-9	5-methylhexyl 3-methylbutanoate	Z23	0.43 ± 0.05	5.66 ± 1.45	12.52 ± 3	24.1 ± 1.17	5.49 ± 0.08	0.63 ± 0.28	44.15 ± 8.19	1.43 ± 0.22	2.66 ± 0.4	1 ± 0.07	38.21 ± 1.11	0.92 ± 0.62
22.776	35852-42-7	4-methylpentyl 4-methylpentanoate	Z24	0.73 ± 0.09	0.07 ± 0.05	3.71 ± 0.96	3.15 ± 0.04	11.02 ± 0.39	3.61 ± 0.37	0.41 ± 0.31	3.97 ± 0.11	7.79 ± 0.73	3.03 ± 0.29	0.72 ± 0.17	6.04 ± 0.35
23.067	110-42-9	decanoic acid, methyl ester	Z25	0.19 ± 0.01	0.03 ± 0.01	0.04 ± 0	0.4 ± 0.02	0.02 ± 0	0.01 ± 0	0.06 ± 0.01	0.1 ± 0.02	0.03 ± 0.03	0.05 ± 0.01	0.05 ± 0.01	0.08 ± 0.02
23.65	56423-43-9	butanoic acid, 3-methyl-, heptyl ester	Z26	0.09 ± 0.05	4.56 ± 1.36	1.73 ± 0.45	3.24 ± 0.03	0.13 ± 0.01	21.51 ± 0.84	20.64 ± 1.77	0.26 ± 0.02	0.06 ± 0	23.75 ± 0.5	22.96 ± 0.26	0.26 ± 0.03
23.973	6378-65-0	hexanoic acid, hexyl ester	Z27	5.8 ± 0.91	0.35 ± 0.11	26.38 ± 8.55	7.51 ± 0.32	29.15 ± 0.19	6.2 ± 0.65	2.47 ± 0.17	0.72 ± 0.15	13.43 ± 2.48	5.05 ± 0.83	38.46 ± 0.99	14.02 ± 0.04
24.952	1215128-05-4	6-Methyl-4-heptenyl 2-methylbutanoate	Z28	0.04 ± 0.02	4.29 ± 1.44	2.17 ± 0.44	42.97 ± 1.29	-	26.42 ± 1.02	12.41 ± 1.05	61.25 ± 0.45	0.04 ± 0.02	27.27 ± 2.02	13.67 ± 0.13	57.69 ± 0.88
25.183	1215128-06-5	6-methylhept-4-en-1-yl 3-methylbutanoate	Z29	0.05 ± 0.02	25.18 ± 6.08	12.14 ± 2.95	21.43 ± 0.29	0.07 ± 0.02	31.05 ± 24.59	84.96 ± 6.09	2.53 ± 0.41	0.03 ± 0.02	74.4 ± 6.64	90.66 ± 1.46	3.95 ± 0.55
25.207	53398-86-0	E-hexanoic acid, 2-hexenyl ester, (e)-	Z30	1.16 ± 0.16	0.03 ± 0	0.28 ± 0.12	1.49 ± 0.03	0.19 ± 0.06	0.32 ± 0.01	1.42 ± 0.15	1.21 ± 0.38	0.11 ± 0.05	0.27 ± 0.02	1.2 ± 0.01	1.4 ± 0.03
25.448	10361-39-4	pentanoic acid, phenylmethyl ester	Z31	2.09 ± 0.27	9.57 ± 1.02	5.63 ± 1.14	10.15 ± 0.33	13.73 ± 0.48	43.8 ± 0.34	43.31 ± 14.27	-	7.87 ± 0.74	40.67 ± 6.15	53.55 ± 1.96	0.02 ± 0.01
29.885		2-methylbutyl 8-methylnon-6-enoate	Z32	-	0.1 ± 0.03	4.47 ± 1.49	17.61 ± 0.87	0.03 ± 0.01	13.76 ± 0.72	46.53 ± 3.23	1.88 ± 0.14	-	10.55 ± 1.38	59.14 ± 0.69	43.77 ± 0.01
29.971	6789-88-4	benzoic acid, hexyl ester	Z33	0.18 ± 0.02	1.42 ± 0.16	11.6 ± 2.99	1.31 ± 0.06	1.53 ± 0.06	14.36 ± 0.69	41.5 ± 10.65	2.69 ± 1.46	0.8 ± 0.14	16.95 ± 1.69	50.58 ± 3.68	14.53 ± 0.78
31.286	6846-50-0	2,2,4-trimethyl-1,3-pentanediol diisobutyrate	Z34	0.49 ± 0.03	0.66 ± 0.03	0.91 ± 0.01	0.46 ± 0	0.35 ± 0.04	0.3 ± 0.03	0.5 ± 0.04	0.27 ± 0.21	0.39 ± 0.02	0.24 ± 0.01	0.45 ± 0.03	0.03 ± 0.02
31.381	1215128-16-7	isopentyl 8-methylnon-6-enoate	Z35	0.42 ± 0.06	0.11 ± 0.05	2.97 ± 0.92	37 ± 0.42	8.88 ± 0.36	3.88 ± 0.25	0.82 ± 0.84	0.64 ± 0.22	3.3 ± 0.68	3.73 ± 0.36	0.34 ± 0.01	0.15 ± 0.15
31.91	2306-91-4	pentadecanoic acid, 3-methylbutyl ester	Z36	0.78 ± 0.12	0.09 ± 0.05	0.46 ± 0.12	36.84 ± 2.2	10.23 ± 0.25	1.6 ± 0.09	7.99 ± 2.53	0.04 ± 0.01	3.87 ± 0.86	1.33 ± 0.14	6.73 ± 1.87	0.12 ± 0.03
31.975	68067-33-4	decanoic acid, 2-methylbutyl ester	Z37	-	-	0.05 ± 0	7.15 ± 1.37	0.1 ± 0.01	0.39 ± 0.15	3.28 ± 0.58	0.38 ± 0.26	0.03 ± 0.01	0.23 ± 0.06	3.26 ± 0.21	-
34.196	1215128-18-9	4-methylpentyl 8-methylnon-6-enoate	Z38	0.25 ± 0.03	0.29 ± 0.12	4.83 ± 1.44	28.41 ± 1.34	5.53 ± 0.26	10.2 ± 0.43	33.82 ± 3.4	0.07 ± 0.01	1.57 ± 0.43	11.71 ± 0.76	5.05 ± 0.08	0.63 ± 0.96
34.72	1215127-97-1	4-methylpentyl 8-methylnonanoate	Z39	-	-	-	19.92 ± 1.72	3.39 ± 0.14	3.09 ± 0.17	9.99 ± 0.92	94.87 ± 1.27	-	3.9 ± 0.42	14.84 ± 0.76	82.08 ± 6.96
35.547	85554-69-4	Z-decanoic acid,3-hexenyl ester	Z40	0.03 ± 0	0.02 ± 0	0.04 ± 0.01	5.99 ± 0.67	0.16 ± 0.01	0.53 ± 0.07	0.41 ± 0.12	23.96 ± 6.12	0.05 ± 0.01	0.88 ± 0.14	0.6 ± 0.05	25.66 ± 3.08
35.666	10448-26-7	Decanoic acid, hexyl ester	Z41	0.07 ± 0.01	0.02 ± 0.01	0.15 ± 0.03	6.5 ± 0.53	0.73 ± 0.08	0.18 ± 0.05	0.57 ± 0.11	18.57 ± 0.69	0.24 ± 0.12	0.22 ± 0.13	0.95 ± 0.08	17.02 ± 1.32
36.86	5129-59-9	methyl 13-methyltetradecanoate	Z42	0.24 ± 0.02	0.03 ± 0	0.04 ± 0.02	0.94 ± 0.17	0.03 ± 0.01	0.05 ± 0.02	0.12 ± 0.04	0.71 ± 0.16	0.01 ± 0	0.05 ± 0.01	0.07 ± 0.01	0.46 ± 0.05
37.468	6471-66-5	nonanoic acid, phenylmethyl ester	Z43	-	-	-	0.1 ± 0.02	0.38 ± 0.02	0.02 ± 0	0.07 ± 0.02	6.55 ± 0.25	0.19 ± 0.07	0.02 ± 0.01	0.09 ± 0.01	6.9 ± 1.18
37.485	60160-17-0	n-capric acid n-heptyl ester	Z44	-	-	0.01 ± 0	0.38 ± 0.06	0.08 ± 0.01	0.03 ± 0	0.11 ± 0.03	4.81 ± 0.27	0.02 ± 0.01	0.04 ± 0.01	0.2 ± 0.01	4.39 ± 0.53
37.75		z-11(13-methyl)tetradecen-1-ol acetate	Z45	-	0.04 ± 0.01	0.09 ± 0.03	0.36 ± 0.03	-	14.54 ± 0.61	4.85 ± 1.16	2.11 ± 0.09	-	20.58 ± 0.48	7.45 ± 0.92	1.87 ± 0.19
38.465		2-hexyldecyl acetate	Z46	0.02 ± 0.01	0.03 ± 0	-	0.12 ± 0.01	0.02 ± 0.01	5.03 ± 0.18	1.44 ± 0.39	0.78 ± 0.02	0.01 ± 0	7.08 ± 0.16	2.53 ± 0.34	0.65 ± 0.08
39.009	629-58-3	1-pentadecanol acetate	Z47	0.08 ± 0.01	0.02 ± 0.01	0.02 ± 0.01	0.08 ± 0.01	4.94 ± 0.28	1.03 ± 0.05	1.05 ± 0.29	1.99 ± 0.07	2.19 ± 0.42	2.2 ± 0.06	2.11 ± 0.29	2.88 ± 0.28
40.101	42175-41-7	decanoic acid, phenylmethyl ester	Z48	-	-	-	0.19 ± 0.02	0.53 ± 0.05	0.15 ± 0.01	0.39 ± 0.21	7.98 ± 0.34	0.2 ± 0.08	0.13 ± 0.02	0.35 ± 0.06	5.25 ± 0.9
40.398	112-39-0	hexadecanoic acid, methyl ester	Z49	0.16 ± 0.03	0.94 ± 0.12	0.62 ± 0.15	3.74 ± 1.27	-	-	0.02 ± 0.01	-	-	-	-	-
40.659	2306-92-5	decanoic acid, octyl ester	Z50	-	-	-	0.18 ± 0.04	-	-	-	0.54 ± 0.01	-	-	-	0.53 ± 0.07
41.003		z-11,13-dimethyl-11-tetradecen-1-ol acetate	Z51	-	-	-	0.01 ± 0	0.1 ± 0.02	0.8 ± 0.05	0.29 ± 0.12	0.11 ± 0	0.03 ± 0.01	2.74 ± 0.12	1 ± 0.19	0.17 ± 0.02
41.483	629-70-9	1-hexadecanol, acetate	Z52	-	-	-	-	0.13 ± 0.02	0.32 ± 0.03	0.13 ± 0.05	0.16 ± 0	0.11 ± 0.1	0.83 ± 0.04	0.39 ± 0.07	0.18 ± 0.03
42.021	628-97-7	hexadecanoic acid, ethyl ester	Z53	0.02 ± 0.01	-	-	0.07 ± 0.02	0.04 ± 0.01	0.5 ± 0.17	0.09 ± 0.01	0.13 ± 0.01	0.18 ± 0.13	1.77 ± 0.27	0.07 ± 0.01	0.13 ± 0.02
		**Aldehydes**													
2.712	590-86-3	butanal, 3-methyl-	Q1	0.68 ± 0.26	0.37 ± 0.05	0.27 ± 0.07	0.2 ± 0.01	0.56 ± 0.35	0.08 ± 0.05	0.18 ± 0.01	0.21 ± 0.01	0.2 ± 0.15	0.03 ± 0	0.14 ± 0.01	0.08 ± 0.01
4.139	1576-87-0	2-pentenal, (e)-	Q2	0.73 ± 0.07	0.46 ± 0.03	0.71 ± 0.05	0.77 ± 0.04	1.35 ± 0.52	0.22 ± 0.06	1.28 ± 0.28	0.77 ± 0.15	0.56 ± 0.18	0.08 ± 0.03	0.75 ± 0.31	0.29 ± 0.02
5.14	66-25-1	hexanal	Q3	50.32 ± 10.37	80.24 ± 6.91	51.63 ± 13.93	61.4 ± 3.18	0.14 ± 0.08	-	0.08 ± 0.02	0.03 ± 0	0.79 ± 1.11	0.08 ± 0.07	0.19 ± 0.26	0.02 ± 0
6.644	6728-26-3	2-hexenal, (e)-	Q4	25.89 ± 6.25	35.07 ± 3.81	54.99 ± 7.28	42.38 ± 3.81	1.26 ± 0.18	0.68 ± 0.45	22.09 ± 9.25	0.59 ± 0.02	1.2 ± 0.52	4.84 ± 1.55	0.2 ± 0.26	1.27 ± 0.7
8.008	111-71-7	heptanal	Q5	0.54 ± 0.12	0.64 ± 0.04	0.28 ± 0.09	0.41 ± 0.03	0.33 ± 0.14	0.04 ± 0	0.06 ± 0.01	0.05 ± 0	0.25 ± 0.07	0.02 ± 0.01	0.05 ± 0.01	0.04 ± 0
8.343	142-83-6	2,4-hexadienal, (e,e)-	Q6	5.26 ± 1.19	5.67 ± 0.5	3.91 ± 0.13	4.53 ± 0.37	-	-	-	0.03 ± 0	-	-	-	0.02 ± 0
10.091	100-52-7	benzaldehyde	Q7	4.35 ± 1.56	2.19 ± 0.39	2.12 ± 0.42	1.73 ± 0.18	1.63 ± 0.6	1.27 ± 0.83	0.9 ± 0.08	1.05 ± 0.08	1.7 ± 0.36	-	-	3.59 ± 0.6
12.554	36431-60-4	5-ethylcyclopent-1-enecarboxaldehyde	Q8	6.3 ± 0.49	1.98 ± 0.25	2.73 ± 0.15	1.58 ± 0.14	11.8 ± 0.9	0.11 ± 0.04	1.1 ± 0.09	0.54 ± 0.06	4.52 ± 1.39	0.02 ± 0.01	1.14 ± 0.07	0.21 ± 0.08
13.086	122-78-1	benzeneacetaldehyde	Q9	3.63 ± 0.43	2.46 ± 0.92	0.86 ± 0.19	1.18 ± 0.04	6.14 ± 1.17	0.04 ± 0.03	0.53 ± 0.77	5.09 ± 0.31	7.94 ± 0.23	2.58 ± 1.29	1.81 ± 0.9	0.27 ± 0.17
15.349	124-19-6	nonanal	Q10	0.66 ± 0.04	0.53 ± 0.14	0.53 ± 0.03	1.22 ± 0.06	0.7 ± 0.06	-	-	2.17 ± 0.03	0.54 ± 0.08	-	0.63 ± 0.41	-
17.092	557-48-2	2,6-nonadienal, (e,z)-	Q11	5.02 ± 0.16	1.74 ± 0.13	1.75 ± 0.23	2.05 ± 0.09	3.32 ± 0.35	0.05 ± 0.02	0.06 ± 0.07	0.24 ± 0.02	1.16 ± 0.3	0.03 ± 0.01	-	0.2 ± 0.02
17.376	18829-56-6	2-nonenal, (e)-	Q12	6.73 ± 0.7	6.93 ± 0.34	4.61 ± 0.23	3.81 ± 0.28	2.58 ± 0.43	-	-	0.07 ± 0.01	1.31 ± 0.33	-	-	0.09 ± 0
19.394	5910-87-2	2,4-nonadienal, (e,e)-	Q13	8.23 ± 0.72	5.4 ± 0.27	3.2 ± 0.17	0.02 ± 0.01	1.07 ± 0.33	0.3 ± 0.08	0.14 ± 0	0.85 ± 0.04	0.92 ± 0.68	0.28 ± 0.03	0.48 ± 0.55	0.86 ± 0.09
21.659	112-44-7	Undecanal	Q14	0.89 ± 0.09	0.02 ± 0	0.27 ± 0.06	0.07 ± 0.02	0.39 ± 0.02	0.11 ± 0.01	-	0.13 ± 0.01	0.61 ± 0.03	0.1 ± 0	0.18 ± 0.29	0.07 ± 0.06
24.614	13019-16-4	2-octenal, 2-butyl-	Q15	2.8 ± 0.21	7.06 ± 1.57	0.44 ± 0.02	1.54 ± 0.32	0.04 ± 0.01	0.07 ± 0.01	-	0.86 ± 0.26	0.01 ± 0	0.06 ± 0.03	0.1 ± 0.01	0.48 ± 0.3
26.805		9-undecenal, 2,10-dimethyl-	Q16	0.13 ± 0.02	0.15 ± 0.04	1.09 ± 0.3	32.17 ± 3.22	0.18 ± 0.02	0.02 ± 0	-	42.42 ± 0.34	0.17 ± 0.04	0.03 ± 0.01	-	35.42 ± 1.85
31.591	65128-96-3	7-Tetradecenal, (Z)-	Q17	0.03 ± 0.01	-	-	0.13 ± 0.01	3.09 ± 0.38	0.32 ± 0.05	0.96 ± 0.29	8.42 ± 2.46	2.56 ± 0.36	0.34 ± 0.07	0.83 ± 0.03	12.01 ± 3.12
32.079	124-25-4	tetradecanal	Q18	0.16 ± 0.03	0.01 ± 0	0.05 ± 0.01	0.57 ± 0.21	8.87 ± 1.36	5.66 ± 1.31	2.52 ± 0.96	1.65 ± 0.19	12.13 ± 2.26	7.01 ± 1.27	3.05 ± 0.12	0.64 ± 0.62
33.373	58594-45-9	13-octadecenal, (z)-	Q19	0.01 ± 0	0.02 ± 0	-	0.03 ± 0.01	0.02 ± 0	0.02 ± 0	0.01 ± 0	-	0.02 ± 0.01	0.03 ± 0	0.01 ± 0	0.02 ± 0.01
34.356	849830-41-7	Z-11-Pentadecenal	Q20	0.43 ± 0.18	0.01 ± 0	0.07 ± 0.02	0.72 ± 0.22	10.88 ± 0.94	0.22 ± 0.1	1.74 ± 0.61	11.61 ± 0.77	10.95 ± 4.9	0.28 ± 0.06	2.43 ± 0.15	3.04 ± 1.33
34.942	2765/11/9	pentadecanal-	Q21	5.61 ± 1.79	0.06 ± 0.02	0.79 ± 0.37	15.46 ± 4.53	3.67 ± 0.18	19.32 ± 3.58	13.6 ± 3.9	0.02 ± 0	3.91 ± 0.12	21.73 ± 3.55	15.47 ± 0.89	0.07 ± 0.05
36.132	57491-33-5	e-11-hexadecenal	Q22	0.03 ± 0.02	0.05 ± 0.01	0.07 ± 0.04	0.29 ± 0.05	1.14 ± 0.08	14.92 ± 1.28	5.26 ± 1.26	2.55 ± 0.16	0.61 ± 0.27	16.13 ± 2.15	8.35 ± 0.97	2.16 ± 0.29
37.176	56219-04-6	cis-9-Hexadecenal	Q23	0.02 ± 0	0.03 ± 0.01	0.02 ± 0	-	0.43 ± 0.04	0.06 ± 0.01	0.13 ± 0.04	0.17 ± 0.06	0.37 ± 0.13	0.07 ± 0.01	0.19 ± 0	-
37.633	629-80-1	Hexadecanal	Q24	0.02 ± 0	-	-	0.12 ± 0.05	1.34 ± 0.13	1.96 ± 0.32	0.47 ± 0.22	2.92 ± 0.2	1.24 ± 0.43	2.55 ± 0.61	1.24 ± 0.1	1.28 ± 0.21
39.377	56554-35-9	9,17-octadecadienal, (z)-	Q25	-	-	0.01 ± 0	0.03 ± 0.01	8.46 ± 3.51	0.74 ± 0.33	0.31 ± 0.25	11.97 ± 0.77	16.45 ± 4.63	0.28 ± 0.07	0.65 ± 0.19	8.67 ± 0.7
40.217	629-90-3	heptadecanal	Q26	0.02 ± 0	-	-	0.25 ± 0.09	1.51 ± 0.18	0.25 ± 0.04	0.16 ± 0.08	1.53 ± 0.05	2.26 ± 0.83	0.13 ± 0.06	0.28 ± 0.02	1.01 ± 0.07
20.997	3913-81-3	2-decenal, (e)-	Q27	0.71 ± 0.04	0.46 ± 0.05	2.17 ± 0.22	0.21 ± 0.01	0.57 ± 0.05	0.27 ± 0.01	0.06 ± 0.01	0.15 ± 0.01	0.42 ± 0.22	0.41 ± 0.33	0.23 ± 0	3.02 ± 0.91
		**Alcohols**													
4.447	1576-95-0	2-penten-1-ol, (z)-	C1	1.37 ± 0.32	0.79 ± 0.08	1.05 ± 0.08	1.07 ± 0.09	4.96 ± 0.67	1.73 ± 0.45	1.75 ± 0.26	2.54 ± 0.24	4.16 ± 1.82	0.76 ± 0.15	1.54 ± 0.1	2.1 ± 0.1
6.11	626-89-1	1-pentanol, 4-methyl-	C2	3.21 ± 0.5	6.36 ± 0.89	3.92 ± 0.17	9.77 ± 0.92	34.3 ± 1.21	50.99 ± 4.65	16.22 ± 1.17	25.43 ± 0.65	28.11 ± 6.28	62.7 ± 17.28	26.34 ± 0.83	33.38 ± 1.52
7.012	111-27-3	1-hexanol	C3	7.6 ± 0.11	12.87 ± 1.01	16.02 ± 0.4	19.99 ± 0.71	19.61 ± 4.37	6.77 ± 0.43	5.74 ± 0.97	15.37 ± 0.93	24.93 ± 7.27	11.09 ± 2.21	8.17 ± 0.59	18.34 ± 0.85
12.172	855901-81-4	(E)-6-Methylhept-4-en-1-ol	C4	0.12 ± 0.06	0.06 ± 0.01	0.06 ± 0.02	0.66 ± 0.31	8.45 ± 1.1	-	8.79 ± 2.24	-	9.43 ± 0.9	-	-	-
12.825	100-51-6	benzyl alcohol	C5	-	-	0.02 ± 0.01	-	5.45 ± 1.76	-	0.02 ± 0	-	3.92 ± 1.59	-	-	-
23.595	21592-95-0	4,4,6-trimethyl-cyclohex-2-en-1-ol	C6	4.66 ± 0.27	1.71 ± 0.54	1.96 ± 0.49	3.21 ± 0.18	1.89 ± 0.17	-	0.19 ± 0.01	1.57 ± 0.04	2.5 ± 0.94	0.3 ± 0.29	-	0.98 ± 0.06
26.09		10-methyltricyclo [4.3.1.1(2,5)]undecan-10-ol	C7	0.99 ± 0.04	1.69 ± 0.22	0.26 ± 0.08	0.61 ± 0.1	0.03 ± 0.01	-	-	0.01 ± 0	0.03 ± 0.01	0.02 ± 0	-	-
27.046	26993-32-8	e-2-hexadecacen-1-ol	C8	10.51 ± 1.45	13.28 ± 5.04	26.01 ± 7.9	12.11 ± 1.1	37.04 ± 3.14	0.17 ± 0.11	0.06 ± 0	-	12.87 ± 0.75	0.1 ± 0	0.05 ± 0	-
31.069	80625-44-1	E-11,13-tetradecadien-1-ol	C9	-	-	0.02 ± 0	-	24.58 ± 1.06	-	-	44.13 ± 3.14	24.16 ± 2.14	-	6.18 ± 0.32	25.24 ± 5.83
37.754		e-11(13-methyl)tetradecen-1-ol	C10	-	0.04 ± 0.01	0.05 ± 0.03	0.58 ± 0.07	-	9.26 ± 0.39	3.05 ± 0.74	3.49 ± 0.16	-	13.22 ± 0.14	4.71 ± 0.6	2.93 ± 0.34
		**hydrocarbons**													
28.682	629-62-9	pentadecane	TH1	0.14 ± 0.01	0.05 ± 0.01	0.03 ± 0	-	-	0.06 ± 0.01	0.52 ± 0.09	51.21 ± 9.43	7.96 ± 0.51	0.05 ± 0.04	0.75 ± 0.22	23.49 ± 4.27
30.651	1560-93-6	pentadecane, 2-methyl-	TH2	16.12 ± 2.32	1.57 ± 0.59	1.8 ± 0.66	6.79 ± 0.59	23.47 ± 2.23	0.59 ± 0.06	20.66 ± 0.42	0.03 ± 0	11.3 ± 0.9	8.2 ± 13.13	0.3 ± 0.03	19.98 ± 0.58
30.84	2882-96-4	pentadecane, 3-methyl-	TH3	2.74 ± 0.43	0.58 ± 0.31	1.3 ± 0.49	1.49 ± 0.18	4.83 ± 0.57	8.91 ± 1.44	12.89 ± 1.91	4.48 ± 0.23	1.78 ± 0.11	6.63 ± 1.09	18.09 ± 1.08	5.57 ± 0.42
33.503	1560-92-5	hexadecane, 2-methyl-	TH4	7.15 ± 0.94	1.1 ± 0.28	1.87 ± 0.69	3.18 ± 0.38	10.92 ± 1.36	24.54 ± 2.85	21.18 ± 2.09	9.49 ± 0.44	4.36 ± 0.18	20.67 ± 2.46	29.85 ± 1.68	13.15 ± 1.02
34.53	629-78-7	heptadecane	TH5	19.54 ± 2.07	0.47 ± 0.11	2.01 ± 0.75	5.1 ± 0.51	15.76 ± 1.29	16.11 ± 1.92	16.5 ± 2.25	9.93 ± 0.46	9.82 ± 0.21	15.54 ± 1.39	28.28 ± 1.42	12 ± 1.08
36.223	1560-89-0	heptadecane, 2-methyl-	TH6	0.33 ± 0.04	0.03 ± 0.01	0.05 ± 0.01	0.09 ± 0.01	0.5 ± 0.07	0.15 ± 0.02	0.29 ± 0.05	0.34 ± 0.02	0.25 ± 0.05	0.16 ± 0.02	0.47 ± 0.03	0.56 ± 0.05
36.424	6418-44-6	heptadecane, 3-methyl-	TH7	0.07 ± 0.01	0.04 ± 0.02	0.02 ± 0.01	0.02 ± 0	0.14 ± 0.01	0.16 ± 0.02	0.13 ± 0.02	0.09 ± 0	0.06 ± 0.01	0.18 ± 0.01	0.26 ± 0.02	0.1 ± 0.01
37.2	593-45-3	octadecane	TH8	0.09 ± 0.01	0.14 ± 0.06	0.07 ± 0.01	0.2 ± 0.02	0.46 ± 0.02	0.41 ± 0.06	0.5 ± 0.12	0.69 ± 0.05	0.27 ± 0.02	0.55 ± 0.05	1 ± 0.07	0.66 ± 0.06
39.749	629-92-5	nonadecane	TH9	0.33 ± 0.05	0.02 ± 0.01	0.06 ± 0.02	0.87 ± 0.16	0.34 ± 0.04	0.62 ± 0.03	0.42 ± 0.12	2.07 ± 0.08	0.21 ± 0.04	0.76 ± 0.05	0.9 ± 0.09	2.22 ± 0.21
22.29	629-50-5	tridecane	TH10	0.15 ± 0.03	0.11 ± 0.08	0.11 ± 0.02	0.07 ± 0	0.28 ± 0.04	0.82 ± 0.28	0.94 ± 0.26	0.37 ± 0.05	0.19 ± 0.01	0.52 ± 0.09	0.8 ± 0.03	0.14 ± 0.02
24.363	1560-96-9	tridecane, 2-methyl-	TH11	27.86 ± 3.11	0.65 ± 0.18	2.27 ± 0.74	1.61 ± 2.71	0.1 ± 0	-	0.01 ± 0	-	0.21 ± 0.01	0.02 ± 0.01	0.01 ± 0	0.03 ± 0
25.609	629-59-4	tetradecane	TH12	1.6 ± 0.28	3.45 ± 2.75	2.12 ± 0.63	0.3 ± 0.25	12.45 ± 1.22	28.43 ± 3.2	32.45 ± 5.78	0.12 ± 0.01	5.33 ± 0.12	23.6 ± 2.32	34.31 ± 1.4	1.45 ± 2.31
27.614	1560-95-8	tetradecane, 2-methyl-	TH13	14.61 ± 2.09	8.21 ± 3.37	10.8 ± 3.21	0.52 ± 0.39	44.71 ± 4.37	0.03 ± 0.02	-	0.24 ± 0.04	16.62 ± 1.38	0.06 ± 0.04	-	0.79 ± 0.5
28.11	295-48-7	cyclopentadecane	TH14	5.92 ± 0.8	0.08 ± 0.02	5.23 ± 1.59	5.23 ± 0.13	6.48 ± 0.31	0.37 ± 0.26	0.02 ± 0	0.51 ± 0.1	4.26 ± 0.26	1.97 ± 2.4	0.03 ± 0.02	0.02 ± 0.01
31.721	544-76-3	hexadecane	TH15	5.87 ± 0.9	2.83 ± 1.26	3.03 ± 0.97	6.46 ± 0.51	16.05 ± 1.38	57.15 ± 6.75	41.8 ± 4.28	2.15 ± 0.73	6.23 ± 0.3	49.86 ± 6.56	57.18 ± 2.19	17.77 ± 1.85
30.073	29833-69-0	1-pentadecene, 2-methyl-	TH16	1.86 ± 0.29	-	0.01 ± 0	1.85 ± 0.13	3.2 ± 0.26	0.12 ± 0.01	1.11 ± 0.18	2.66 ± 1.7	1.33 ± 0.1	0.21 ± 0.03	1.44 ± 0.1	5.97 ± 0.38
31.18	35507-10-9	z-8-hexadecene	TH17	0.11 ± 0.01	-	0.04 ± 0.02	2.54 ± 0.15	0.53 ± 0.04	0.4 ± 0.04	0.87 ± 0.15	2.45 ± 0.15	0.22 ± 0.1	0.51 ± 0.07	1.32 ± 0.08	4.49 ± 0.39
32.959		2-methyl-e-7-hexadecene	TH18	0.63 ± 0.09	0.2 ± 0.06	1.57 ± 0.6	2.75 ± 0.21	1.9 ± 0.26	8.49 ± 1.24	13.76 ± 1.41	1.98 ± 0.04	0.5 ± 0.02	6.54 ± 1.07	16.78 ± 0.99	14.13 ± 1.19
33.911	62026-26-0	3-heptadecene, (z)-	TH19	0.9 ± 0.13	0.03 ± 0.01	0.14 ± 0.02	1.82 ± 0.3	14.97 ± 0.47	12.29 ± 1.78	7.61 ± 1.93	21.83 ± 0.81	8.77 ± 4.38	14.51 ± 2.44	10.2 ± 0.41	17.78 ± 1.9
39.173	418766-86-6	z-5-nonadecene	TH20	0.09 ± 0.01	-	-	-	0.06 ± 0.01	-	-	0.04 ± 0.02	0.04 ± 0.01	0.01 ± 0	0.02 ± 0.01	0.04 ± 0.02
		**Terpenoids**													
23.854	17699-14-8	.alpha.-cubebene	T1	0.03 ± 0	1.94 ± 0.69	10.86 ± 2.93	16.5 ± 0.94	0.01 ± 0	12.53 ± 3.36	25.66 ± 2.96	0.46 ± 0.05	-	18.09 ± 1.95	24.9 ± 0.14	0.7 ± 0.34
27.169	3853-83-6	1h-benzocycloheptene, 2,4a,5,6,7,8,9,9a-octahydro-3,5,5-trimethyl-9-methylene-, (4as-cis)-	T2	1.69 ± 0.03	1.25 ± 0.17	7.97 ± 2.23	13.17 ± 0.27	3.35 ± 0.11	3.58 ± 0.81	5.63 ± 1.36	0.52 ± 0.02	1.25 ± 0.11	0.45 ± 0.03	3.12 ± 2.27	0.32 ± 0.25
27.213	28973-97-9	cis-.beta.-farnesene	T3	1.73 ± 0.43	1.18 ± 0.51	12.74 ± 3.75	3.59 ± 0.05	2.64 ± 0.21	81.02 ± 7.41	89.16 ± 18.54	30 ± 8.66	2.08 ± 0.6	61.08 ± 13.96	111.66 ± 4.92	1.78 ± 0.27
28.181	60909-27-5	himachala-2,4-diene	T4	0.15 ± 0.02	0.14 ± 0.04	31.01 ± 0.71	92.16 ± 1.52	0.62 ± 0.02	63.77 ± 4.95	9.47 ± 4.69	0.6 ± 0.68	0.22 ± 0.02	45.71 ± 12.05	7.12 ± 3.95	7.73 ± 0.38
28.4	18252-46-5	cis-.alpha.-bergamotene	T5	0.2 ± 0.05	-	0.89 ± 0.06	0.04 ± 0.03	0.51 ± 0.03	0.02 ± 0	-	8.13 ± 2.25	0.36 ± 0.12	0.02 ± 0	0.02 ± 0	0.11 ± 0.07
28.642	10208-80-7	.alpha.-muurolene	T6	-	0.29 ± 0.08	2.28 ± 0.56	1.98 ± 0.62	-	2.33 ± 0.23	3.33 ± 2.29	0.57 ± 0.15	-	1.79 ± 0.24	2.98 ± 2.27	0.03 ± 0
29.012	78204-62-3	.alpha.-dehydro-ar-himachalene	T7	0.22 ± 0.01	0.26 ± 0.01	1.02 ± 0.13	1.66 ± 0.09	0.33 ± 0.01	0.52 ± 0.01	0.11 ± 0	-	0.13 ± 0.03	0.43 ± 0.06	0.07 ± 0	-
29.472	51766-65-5	.gamma.-dehydro-ar-himachalene	T8	0.5 ± 0.03	0.58 ± 0.02	1.98 ± 0.21	2.16 ± 0.06	0.48 ± 0.02	0.55 ± 0.47	0.02 ± 0	-	0.17 ± 0.04	0.73 ± 0.09	0.01 ± 0	-
29.729	19419-67-1	ar-himachalene	T9	0.44 ± 0.04	0.56 ± 0.11	5.3 ± 1.2	8.95 ± 0.19	0.76 ± 0.06	2.23 ± 0.08	0.1 ± 0.06	-	0.33 ± 0.06	1.86 ± 0.32	0.06 ± 0.01	-
29.957	21391-99-1	.alpha.-calacorene	T10	0.03 ± 0	0.63 ± 0.2	4.06 ± 1.11	6.84 ± 0.17	0.12 ± 0.01	9.48 ± 0.83	13.68 ± 1.5	-	0.06 ± 0.01	6.64 ± 1.8	15.25 ± 0.74	-
30.547	7212-44-4	3,7,11-trimethyl-1,6,10-dodecatrien-3-ol	T11	3.87 ± 0.41	0.05 ± 0.01	0.02 ± 0	1.08 ± 0.03	5.89 ± 0.45	0.41 ± 0.05	0.09 ± 0.02	-	6.6 ± 1.38	0.71 ± 0.23	0.14 ± 0.04	0.04 ± 0.03
30.554	142-50-7	nerolidol	T12	1.07 ± 0.15	0.02 ± 0	0.04 ± 0	0.44 ± 0.02	1.94 ± 0.11	0.23 ± 0.01	0.16 ± 0.04	-	1.85 ± 0.39	0.29 ± 0.01	0.22 ± 0.02	0.01 ± 0
33.62	19435-77-9	(4ar,5r,9ar)-1,1,4a,8-tetramethyl-2,3,4,4a,5,6,7,9a-octahydro-1h-benzo [7]annulen-5-ol	T13	0.04 ± 0	0.03 ± 0	0.06 ± 0.01	0.17 ± 0.01	0.08 ± 0.01	0.19 ± 0.01	0.16 ± 0.04	0.21 ± 0.1	0.04 ± 0.01	0.29 ± 0	0.21 ± 0.02	0.18 ± 0.14
19.473	432-25-7	1-cyclohexene-1-carboxaldehyde,2,6,6-trimethyl-	T14	2.66 ± 0.15	1.54 ± 0.12	2.05 ± 0.08	0.02 ± 0	0.31 ± 0.05	-	-	-	0.4 ± 0.11	-	-	-
22.125	25152-84-5	2,4-decadienal, (e,e)-	T15	4.77 ± 0.15	5.06 ± 0.49	7.2 ± 2	5.5 ± 0.24	16.46 ± 3.37	0.49 ± 0.71	0.1 ± 0	0.05 ± 0	9.95 ± 6.63	0.72 ± 0.05	0.1 ± 0.02	0.07 ± 0.01
15.15	78-70-6	linalool	T16	4.38 ± 0.84	2.25 ± 0.39	0.87 ± 0.17	3.44 ± 0.06	5.42 ± 0.21	4.68 ± 0.97	2.11 ± 1.09	6.06 ± 0.38	2.5 ± 0.38	4.25 ± 0.2	1.91 ± 0.28	5 ± 0.22
18.695	10482-56-1	l-.alpha.-terpineol	T17	0.61 ± 0.13	-	-	-	0.06 ± 0.01	-	0.07 ± 0.03	0.08 ± 0.01	0.11 ± 0.03	0.02 ± 0	-	-
26.614	17283-81-7	2-butanone,4-(2,6,6-trimethyl-1-cyclohexen-1-yl)-	T18	0.17 ± 0.02	0.24 ± 0.02	0.47 ± 0.07	0.26 ± 0.02	0.46 ± 0.01	1.34 ± 0.1	0.64 ± 0.05	0.22 ± 0.01	0.25 ± 0.03	1.57 ± 0.09	0.61 ± 0.24	0.47 ± 0.22
18.121	562-74-3	terpinen-4-ol	T19	0.32 ± 0.07	0.61 ± 0.09	2.07 ± 0.22	-	0.05 ± 0	-	-	-	0.17 ± 0.03	-	-	-
26.267	127-41-3	α-Ionone	T20	1.54 ± 0.05	4.8 ± 0.31	4.34 ± 0.46	5.83 ± 0.12	6.3 ± 0.11	27.3 ± 1.42	6.32 ± 0.24	0.39 ± 0.08	4.12 ± 0.29	30.53 ± 1.31	9.5 ± 0.43	0.95 ± 0.38
		**ketones**													
14.807	821-55-6	2-nonanone	K1	0.7 ± 0.08	0.05 ± 0	0.55 ± 0.06	0.08 ± 0	0.05 ± 0.02	0.73 ± 0.25	1.14 ± 0.05	0.14 ± 0.01	0.42 ± 0.57	0.29 ± 0.13	0.58 ± 0.05	0.19 ± 0.19
22.011	112-12-9	2-undecanone	K2	0.1 ± 0.02	0.69 ± 0.16	0.77 ± 0.14	4.86 ± 0.19	0.42 ± 0.03	6.89 ± 0.19	5.36 ± 0.34	11.18 ± 0.2	0.18 ± 0.02	3.36 ± 3.88	5.82 ± 0.14	9.38 ± 0.64
38.24	502-69-2	2-pentadecanone, 6,10,14-trimethyl-	K3	0.04 ± 0.01	-	-	0.07 ± 0.01	0.03 ± 0	-	-	0.17 ± 0.01	0.02 ± 0.01	-	-	0.14 ± 0.03
		**Acids**													
12.008	111-14-8	heptanoic acid	S1	0.02 ± 0	0.02 ± 0	0.04 ± 0	0.11 ± 0.01	0.06 ± 0.03	0.34 ± 0.02	0.97 ± 0.09	1.08 ± 0.13	0.06 ± 0.02	0.24 ± 0.05	1.71 ± 0.18	1.75 ± 0.22
14.522	13419-69-7	2-hexenoic acid, (e)-	S2	0.81 ± 0.03	0.02 ± 0.02	0.11 ± 0.01	-	8.92 ± 4.22	-	0.06 ± 0.01	-	4.13 ± 2.46	0.02 ± 0	9.77 ± 0.59	0.81 ± 0.86
23.32	59320-77-3	8-methyl-6-nonenoic acid	S3	-	0.12 ± 0.11	-	0.18 ± 0.16	1.31 ± 1.09	0.21 ± 0.07	0.48 ± 0.39	0.02 ± 0.01	5.47 ± 3.43	0.28 ± 0.11	0.57 ± 0.47	0.15 ± 0.12
		**phenols**													
14.574	90-05-1	phenol, 2-methoxy-	F1	-	-	-	-	-	0.15 ± 0.06	-	-	42.4 ± 11.57	0.06 ± 0.04	0.01 ± 0	0.09 ± 0.04
28.927	96-76-4	2,4-di-tert-butylphenol	F2	0.76 ± 0.13	1.49 ± 0.25	1.93 ± 0.22	1.84 ± 0.49	0.69 ± 0.24	0.53 ± 0.16	0.45 ± 0.49	-	0.73 ± 0.48	0.43 ± 0.02	-	-
22.845	31143-55-2	phenol, 3-methyl-6-propyl-	F3	3.25 ± 1.51	3.19 ± 1.28	2.05 ± 0.51	1.19 ± 0.06	0.52 ± 0.01	-	0.41 ± 0.15	-	0.92 ± 0.78	-	0.31 ± 0.07	-
		**others**													
17.536	24168-70-5	2-methoxy-3-(1-methylpropyl)-pyrazine	O1	0.57 ± 0.03	0.18 ± 0.02	0.12 ± 0	0.11 ± 0	0.06 ± 0.01	0.01 ± 0	-	0.01 ± 0	0.02 ± 0	-	-	-
17.897	24683-00-9	2-methoxy-3-(2-methylpropyl)-pyrazine	O2	28.54 ± 1.95	19.48 ± 1.35	17.03 ± 0.82	8.21 ± 0.3	12.88 ± 0.98	1.88 ± 0.13	2.36 ± 0.16	3.88 ± 0.3	14.9 ± 0.7	0.82 ± 0.19	0.9 ± 0.09	2.8 ± 0.26
3.199	3208-16-0	furan, 2-ethyl-	O3	5.49 ± 1.24	3.37 ± 0.07	3.41 ± 1.44	4.93 ± 0.36	2.09 ± 0.3	0.42 ± 0.12	1.43 ± 0.1	2.48 ± 0.31	0.87 ± 0.57	0.16 ± 0.02	0.48 ± 0.1	0.78 ± 0.03
11.063	3777-69-3	furan, 2-pentyl-	O4	28.12 ± 5.44	19.99 ± 0.56	26.39 ± 4.43	26.46 ± 1.69	17.32 ± 3.97	0.95 ± 0.22	4.21 ± 0.64	5.54 ± 0.56	17.47 ± 4.02	0.63 ± 0.03	2.34 ± 0.42	3.09 ± 0.12
29.592	17092-92-1	2(4h)-benzofuranone, 5,6,7,7a-tetrahydro-4,4,7a-trimethyl-, (r)-	O5	0.76 ± 0.1	0.18 ± 0.06	0.47 ± 0.08	0.49 ± 0.06	0.57 ± 0.07	0.25 ± 0.02	0.17 ± 0.12	-	0.63 ± 0.11	1.58 ± 0.08	0.58 ± 0.39	-
33.086	27198-63-6	14-methyl-oxacyclotetradecan-2-one	O6	0.71 ± 0.05	0.16 ± 0.02	4.07 ± 0.9	1.11 ± 0.01	1.35 ± 0.03	1.34 ± 0.08	4.88 ± 0.87	11.22 ± 1.42	1.17 ± 0.18	1.23 ± 0.09	6.44 ± 0.38	0.03 ± 0.02
37.892	106-02-5	oxacyclohexadecan-2-one	O7	0.06 ± 0	-	0.18 ± 0.05	0.05 ± 0	0.24 ± 0.01	0.09 ± 0.01	0.58 ± 0.14	0.44 ± 0.03	0.19 ± 0.03	0.15 ± 0.08	0.88 ± 0.11	0.42 ± 0.06
24.841	61142-64-1	1,2-oxaborole,2,3,4-triethyl-2,5-dihydro-5,5-dimethyl-	O8	1.78 ± 0.14	1.68 ± 0.2	0.9 ± 0.07	0.78 ± 0.07	0.17 ± 0.03	5.21 ± 0.06	1.9 ± 0.2	0.64 ± 0.07	0.33 ± 0.05	8.85 ± 0.27	2.12 ± 0.1	1.13 ± 0.17
26.506	56691-74-8	(2,6,6-trimethylcyclohex-1-enylmethanesulfonyl)benzene	O9	4.33 ± 0.13	7.18 ± 0.53	1.06 ± 0.11	3.53 ± 0.46	0.09 ± 0.02	0.02 ± 0	0.03 ± 0.01	-	0.06 ± 0.02	0.03 ± 0	0.02 ± 0.01	-
17.306	4861-58-9	thiophene, 2-pentyl-	O10	0.92 ± 0.24	0.76 ± 0.55	0.26 ± 0.01	0.21 ± 0.03	0.06 ± 0.02	0.29 ± 0.05	0.18 ± 0.02	0.26 ± 0.01	0.12 ± 0.08	0.17 ± 0.05	0.15 ± 0.01	0.25 ± 0.02
		**total relative content of volatiles**		526.49 ± 39.99	663.67 ± 62.31	811.76 ± 66.75	1082.96 ± 57.66	761.66 ± 46.05	917.03 ± 64.86	1070.68 ± 105.84	894.39 ± 10.49	603.49 ± 37.45	931.68 ± 84.19	1210.62 ± 12.19	796.87 ± 34.22

Note: values are means ± standard deviation; “-”: indicates not detected; “The total aroma content of volatiles” was calculated as the sum of all aroma metabolites; RT: retention time.

## Data Availability

The datasets presented in this study can be found in online repositories of the National Center for Biotechnology Information (NCBI). The accession number is PRJNA932246.

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
