# Peer review of "Analysis of Volatile Aroma Components and Regulatory Genes in Different Kinds and Development Stages of Pepper Fruits Based on Non-Targeted Metabolome Combined with Transcriptome"

_ijms, 2023, doi:10.3390/ijms24097901_

Round 1
Reviewer 1 Report
VACs in pepper fruits from different varieties and developmental stages were determined by HS-SPME combined with GC-MS. Moreover, RNA-seq was used to analyze the gene expression of fruits in corresponding varieties and development stages, and the structural genes and co-expression networks related to the synthesis path of aroma volatiles were identified by metabolome-transcriptome association analysis. This research would lay a foundation for further improving the aroma of pepper fruits and promoting the improvement of pepper fruit flavor through gene and metabolic engineering methods. But the current version of this manuscript is not good enough, some issues should be addressed.
1. Why were these four varieties chosen as research materials? It should be explained. Why the varieties from other species, Capsicum frutescens, Capsicum baccatum, Capsicum assamicum, and Capsicum pubescens, are not included in this study?
2. In Line 117, details about 36 samples need be added, including four varieties and three developmental stages. Pictures of pepper fruits from different samples should be showed in this part.
3. Based on the section of “4.2. Untargeted metabolome analysis”, HS-SPME was used to extract the VACs from dried pepper fruit powder, not from fresh samples. However, type and content of VACs are quite different between dried pepper fruit powder and fresh pepper fruit. Please explain the reason.
4. Some statments are not correct and accurate. “The concentration of total VOCs in HDL1 was significantly higher than that of other pepper varieties in the breaking stage...” (Lines 131-133). Based on Figure 1a, there is likely no significant difference between HDL1 and other varieties in the breaking stage. So I greatly suggest authors check the expression of all results to make sure that they are accurate.
5. Based on Figure 1a, there was no significant difference between breaking stage and maturation stage in all four varieties, while significant difference between breaking stage and green stage in all four varieties was showed. Therefore, authors should just focus on transcriptome change of pepper fruits in green stage and breaking stage to figure out the key genes invovled in the essential pathways.
6. Qualitive analysis of VACs was stated as “They then compared with MS in the NIST database to determine the species of VACs”. In my opinion, it is not enough, and retention index (RI) should be calculated using the retention times of n-alkanes that had been injected into the same instrument under the same chromatographic conditions.
Author Response
- 为什么选择这四个品种作为研究材料?应该解释一下。为什么其他物种的品种,辣椒果,辣椒,不在本研究中?
回应:感谢您对本研究的建议。本研究的1个品种被选为研究材料,因为'HDL2'(辣椒)是海南省特有的地方品种,水果闻起来芳香,加工产品海南黄登龙椒酱是海南省最受欢迎和最有前景的特色农产品。我们希望明确其挥发性香气物质的主要成分、含量和变化,并了解其挥发性香气物质是否与其他品种不同。“HDL1”(与HDL1果形不同的杂交品种)是与“HDL1”密切相关的品种,而“GJ”与“HDL1”属于同一栽培品种。此外,中国种植的辣椒品种大多属于辣椒L.和辣椒L.。因此,本研究主要针对辣椒L.和辣椒L.的栽培品种。品种。因此,我们选择了市场上更受欢迎的辣椒品种“CTJ”(辣椒L.)。以上四个品种在遗传关系、栽培品种和品种上都具有代表性。因此,我们重点分析这四个品种。选择“HDL90”的原因已在本文第 92-585 行中简要描述。此外,在文章中进行了补充(593-<>行)。
- 在第 117 行中,需要添加有关 36 个样品的详细信息,包括四个品种和三个发育阶段。这部分应显示来自不同样品的胡椒果实的图片。
答复:文中36个样本的标签已更改(第119行)。4个品种和3个发育阶段的辣椒果实的表型图像如图1所示。
- 根据“2.非靶向代谢组分析“,HS-SPME用于从干胡椒果粉中提取VAC,而不是从新鲜样品中提取VAC。然而,干胡椒果粉和新鲜胡椒果的VAC的类型和含量有很大不同。请解释原因。
响应:本文所表达的“冻干”是指鲜椒果实在液氮中的固态,更好地研磨成粉末,提高提取效率。事实上,我们使用新鲜的样品。在这篇文章中,我们没有表达清楚,已经对606-608行进行了修改。
- 有些说法是不正确和准确的。HDL1中总VOCs的浓度在破碎阶段明显高于其他辣椒品种......“(第131-133行)。根据图1a,HDL1与其他品种在破碎阶段可能没有显着差异。因此,我强烈建议作者检查所有结果的表达,以确保它们是准确的。
回应:得益于老师的精心发现,本研究主要从品种和发育期两个维度分析辣椒果实中的香气挥发物。由于一般鲜椒主要在成熟期食用,因此在分析本研究品种间差异时,我们重点分析了成熟胡椒果实中总VOCs的数据分析。文本中中断阶段的数据出现了冗余错误,这些错误已在文本中更改(第 131-135 行)。已检查其他结果。
- 从图1a可以看出,<>个品种的破龄期和成熟期无显著差异,而<>个品种的破龄期和绿色期差异均有显著差异。因此,作者应只关注辣椒果实在绿化期和破发期的转录组变化,找出基本通路中涉及的关键基因。
回应:在这项研究中,我们的转录组和代谢组结果一致,绿色期和颜色转换期与成熟期之间分别显示出显着差异。因此,我们在分析DEGs时,重点关注绿色期和颜色转换期之间的差异,以及绿色期和成熟期之间的差异。虽然我们也分析了转折期和成熟期的差异基因,但这是因为我们想展示更全面的数据,可以更直观地突出绿色期和转折期、成熟期的区别。
- VAC的定性分析被描述为“然后他们与NIST数据库中的MS进行比较以确定VAC的种类”。在我看来,这还不够,保留指数(RI)应该使用在相同色谱条件下注入同一仪器的正构烷烃的保留时间来计算。
回应:我们比较和审查从NIST数据库中检索到的鉴定结果,以确保手动代谢物鉴定的准确性。值得一提的是,人工审查的标准是我们筛选NIST数据库中相似度超过800点的代谢物。因此,我们没有在表中提供保留指数的信息。
Reviewer 2 Report
In this study HS-SPME-GC-MS combined with transcriptome sequencing are used to analyze the composition and formation mechanism of volatile aroma compounds in different kinds and development stages of pepper fruits. However, there are many problems in the arrangement and composition of the manuscript. For example, the language is simple and repetitive, verbosity, improper paragraph arrangement, and it is not suitable for readers' reading habits. In the abstract: “There were significant qualitative and quantitative differences among different varieties and developmental stages”. “This study revealed the differences of aroma components and contents in pepper fruits of different varieties and development stages...”
Detailed suggestions are as follows:
1. Results section. Figure 10 in this manuscript can be put in front as Figure 1. It can also introduce the experimental sample in detail and its abbreviation, such as GJ, HDL1, HDL2, and CTJ. The abbreviations in the text need to be explained when they first appear.
2. L39-40, Notice how the Latin names of plants are written in the text. Capsicum chinense, C. annuum, C. frutescens..
3. In this manuscript, Figure1 does not clearly show the results of the experiment, but rather creates confusion. Excessive use of tags in the diagram. Statistical analysis is meant to illustrate results, not to show how many methods you used.
4. L 129. “Figure1 Changes in VACs in pepper fruits during different varieties and developmental periods” . There are no VOCs in Figure1
5. In this manuscript, Figure S1, Table S1, Table S3, Table S4 appears as an attachment file. In fact, these are major experimental data that should be included in the text as much as possible.On the contrary, many of the following figures are not the direct result of this study and are unnecessary. The results are many. When writing a paper, it is a question of what should be in the body and what should be in the supplementary. Please Consider Figure 6,7,8,9 carefully.
6. The title of the paper, including the subheadings (2.1, ... 2.5) in the manuscript, is too long and can simplify writing
7. In the Discussion and The Conclusion part, try to write the name of the plant sample, do not recommend to write abbreviations (such as HDL1)
In addition, authors are recommended to have a professional English proofreading service before submission.
Author Response
- Results section. Figure 10 in this manuscript can be put in front as Figure 1. It can also introduce the experimental sample in detail and its abbreviation, such as GJ, HDL1, HDL2, and CTJ. The abbreviations in the text need to be explained when they first appear.
Response: Figure 10 has been placed at the beginning of the article, renumbered as Figure 1 ; annotated, see (lines 110-113).
- L39-40, Notice how the Latin names of plants are written in the text. Capsicum chinense, C. annuum, C. frutescens..
Response: Changes have been made in the text (lines 39-40).
- In this manuscript, Figure1 does not clearly show the results of the experiment, but rather creates confusion. Excessive use of tags in the diagram. Statistical analysis is meant to illustrate results, not to show how many methods you used.
Response: The confusion caused by Figure 1 to you has been simplified and presented in the new combination of Figure 2a. (157-162 Lines)
- “Figure1 Changes in VACs in pepper fruits during different varieties and developmental periods” . There are no VOCs in Figure1
Response: Thanks to the teacher's careful discovery, the y-axis of Figure 1 has been re-annotated and shown in the new combination diagram (Figure 2a).
- In this manuscript, Figure S1, Table S1, Table S3, Table S4 appears as an attachment file. In fact, these are major experimental data that should be included in the text as much as possible.On the contrary, many of the following figures are not the direct result of this study and are unnecessary. The results are many. When writing a paper, it is a question of what should be in the body and what should be in the supplementary. Please Consider Figure 6,7,8,9 carefully.
Response: Figure S1 has been recombined into the c-map and d-map in Figure 2. Table S1 is not shown in the text because of too much data, which will affect the overall length of the article. Tables S3 and S4 are the original data of the experiment, and can not present the most intuitive research results to the readers. Pictures 6,7,8 and 9 are based on the original data sequenced in this study. Through analysis, they can give readers an intuitive reflection of the results of this study, so they are placed in the text.
- The title of the paper, including the subheadings (2.1, ... 2.5) in the manuscript, is too long and can simplify writing
Response: The subheading has been changed accordingly in the text.
- In the Discussion and The Conclusion part, try to write the name of the plant sample, do not recommend to write abbreviations (such as HDL1)
Response: The conclusion part of the abbreviation problem has been modified accordingly. In the discussion part, we believe that the abbreviation can be used to present the results of the study to the readers more intuitively, but we have improved it as follows : 'HDL1' is changed to 'HDL1 ( Hainan Huangdenglong Pepper )'.
- In addition, authors are recommended to have a professional English proofreading service before submission.
Response: English proofreading has been done for this article.
Reviewer 3 Report
IJMS2272750
In this study, an untargeted metabolomics combined with transcriptomics approach was used to study aroma components of pepper fruits during development.
Overall, an interesting descriptive study that will yield important leads for the further understanding of volatile aroma in pepper fruits. However, the message of the manuscript can improve substantially by a much more precise formulation of findings and statements. Next to my suggestions below, I strongly suggest that the authors carefully go through their text to see whether the message comes through.
General remark: there are many typo’s in not having spaces between final character and reference (e.g.) end of sentence[12] or between individual words. Please check carefully editing.
Specific suggestions to consider:
Abstract line 26. Transcription factors were identified, but where these also related to the regulation of volatile emissions? Please clarify.
Introduction. Line 42-45. Pepper fruits contain specific compounds but do not contain health function, such as is now implicated. Instead, the compounds do posses various properties that can be related to health. Also the last sentence in this Alinea: better rephrase as medical, antiseptic .. applications.
Line 53, while triterpenoids are indeed synthesized via the MVA pathway, the final products are, at ambient temperature, not volatile and hence, cannot contribute to aroma. The other terpenoid subclasses named here, are volatile.
Line 55, While indeed terpene synthases (please correct synthetases) are key enzymes that determine the terpenoid profile, both the MVA and MEP pathway produce what is considered to be the direct substrates for TPA, but do not include them.
Line 60, ‘other metabolomic pathways’ is too general and can relate to any pathway not specific for benzenoid biosynthesis.
Line 72, not clear what is meant with ‘the atypical synthetic pathways’.
General suggestion: do not use the term ‘also’ too many times
Line 81, Yucatan, Cachuca and CNPH are not volatile compounds, such as is now referred to in this sentence.
Line 83, ‘but’ suggests a contradiction. However, this sentence is actually giving more details on the timing of emission.
Line 92, and Line 103. The authors intend to profile a ‘ characteristic local variety’, but the selection of the other three varieties was mot motivated.
Results
Line 122, 123 what is meant with ‘the kinds’ ? The biochemical classes, or specific individual metabolites? ‘A relative content of total volatiles’ is contractionary. I guess here is meant: the relative content of specific compounds in the total blend showed significant differences. Please clarify.
Line 137, please specify what is meant with: the first category of volatileS.
Figure 1 In panelA, the explanation of the Y-axis is missing in the figure. The use of both *** and P-values, is confusing. Better use the *** in the figure and explain in the legend what this stand for, ***, P<0.001 etc. What is meant with ‘groups’ on the X-axis? Are these the varieties? And in panel B, what is a, b, c in this grouping? Please adapt the legend accordingly. In legend of biochemical subclasses, all terms are written starting with a Capital, except for ketones.
Line 161, differences were not analysed by relative content, this is not a method of analysis. What is most likely meant here: Differences in varieties and throughout development were analysed using the relative content of each of the metabolites.
Line 164, 165. ‘excellent reproducibility’ suggests a technical replicate, not a biological one. Please clarify what exactly was the experimental set-up.
Figure 2, more detailed information should be given in the legend, including the chosen algorithm to come to this heatmap.
Also in the accompanying text, I suggest to include a description of the main results, including that metabolites derived from specific biochemical pathways do not show obvious clustering, while a clear separation of green stages versus the breaking and maturation stage is visible in the abundances of specific metabolites.
Line 185 ‘were chosen’, ‘three stages of development’.
Line 174-223. I’m a bit surprised by the motivation of the authors to first do PLS-DA analysis and include these in the supplemental data, including a list of discriminant compounds with proper significance (high VIP, low P values, high log2FC), while the main manuscript shows PCA plots, for which no quantitative data for the identified marker metabolites are given. While both analysis are correct in itself, I am curious to the motivation of the authors.
Furthermore, the fact that specific metabolites are deviating from a specific comparison in the PCA plots, is not perse an indicator for being a ‘marker metabolite’ Also, there is no motivation for showing PC1, PC2, and/or PC3 in various combinations. Hence, this seems a bit arbitrary cherry picking. The legends of Figure 3, should also be more neutral. Specifically: what is meant with ‘young stage’ in figure3b right panel?
Line 247/248. The fact that many genes are down-regulated is not perse explanatory for their importance in formation and release of volatile metabolites. This is an overall transcriptome, and includes genes that are related to various processes not related to formation of aroma compounds, but most likely with other aspects of fruit ripening, including many processes that are shuttled down in the final developmental stage. In this stage of the description of the transcriptome analysis, no relations with any biochemical gene expression can be made, and most specifically not with volatile formation at all. Actually, the authors nicely describe such relations in the following section, where they compare go-annotations in various comparisons. Here, it appears that indeed many genes putatively related to the formation of volatile compounds are enriched.
Legend of figure 4 should be more informative, such as describe the 36 samples, in a bit more detail: genotypes / developmental stage…
Line 259, what is meant with ‘the effect of genes (which can be read as treatment?) at transcriptional level?
Figure 5, Please be more detailed in legend descitpion: differentially expressed genes as result of ….? / Venn diagrams indicating DEGs that are overlapping in specific comparisons. Why are some terms in panels g, h, m, highlighted? What is the reason to name the last subpanel ‘m’ instead of ‘i’? This suggest that a draft version of the figure is shown here. Please check.
Line 303, 90/35 what? Genes, metabolites, other?
Figure 6, what is the colour coding bar indicating? Expression level, Z-scores other?
Line 359, differential related to variety or developmental stage?
Line 411, what is used to identify whether a specific gene encodes for a transcription factor. Please indicate (but perhaps this already done in the method section)
Line 415, Furthermore (typo) / formulation of this sentence is unclear, please adapt.
Legend of figure 8 is unclear: what is meant with ….at different spreading time points….; what are the different colours depicted in panel b indicating? Tekst in hexagons is not readable, is there a accompanying supplemental file, in which more details are presented?
In panel a of figure 9, colors are mentioned, but it is not clear to what this refers (most likely figure 8 panelb?). Panel b & c, indicate which subclass of TFs is meant here, not just give colors, what are the bars indicating?? Please inducate what is specifically / additionally shown in panels d & e.
Overall, a nice and interesting discussion section has been written and content-wise I have not much comments. An final overall conclusion statement could be included.
Mehod section.
Line 603, is there additional information on the HDL2 hybrid?, what are the parents of this hybrid, is one of them HDL1? Is there any more detailed (genebank) description of the specific varieties used? At least, geographical indicators, or were fruits just bought at local markets?
Line 606, what were the conditions in the plant growing chambers?
Line 617-626.
Please provide more details on the metabolome analysis. Has the risk of losing specific (very volatile) metabolites due to the freeze-drying procedure being validated? Please provide details (reference, in supplemental). What was the concentration of the ethanol (70%, 96%, other) and how much volaum was added. What is the volume and the provider of the headspace bottles used. Why were samples incubated at such a high temperature, is this representative for smelling pepper fruits? I would expect either an ambient 20-25C or an oral-environment 37C incubation. Please motify.
Line 624, unclear formuation. I suspect that a PDMS fiber was inserted into the headspace of the extracts. Were indeed individual samples incubated for 60 min? Would you not expect a high amount of degradation products due to the extraction and incubation, instead of representative for the composition of the fruits themselves? Has this method been validated? In my lab, we usually incubate SPME fibers for 10-20 min max to avoid that we actually trap other compound than intended.
Line 640, where QC samples actually included in the data analysis, are these shown in the supplementary data?
Line 644, what is meant with substances? Better use ‘mass spectra’ and metabolite indentation.
Section 5, Conclusion. Actually, I would have expected to read this concluding section at eh end of the discussion section.
Author Response
- General remark: there are many typo’s in not having spaces between final character and reference (e.g.) end of sentence[12] or between individual words. Please check carefully editing.
Response: Thanks to the teacher 's careful discovery, such problems have been modified in the text.
- Abstract line 26. Transcription factors were identified, but where these also related to the regulation of volatile emissions? Please clarify.
Response: Thanks to the teacher's suggestion here, we had hoped to express that these transcription factors may be involved in regulating the production of aroma volatiles. Changed in text (line 25-26)
- Line 42-45. Pepper fruits contain specific compounds but do not contain health function, such as is now implicated. Instead, the compounds do posses various properties that can be related to health. Also the last sentence in this Alinea: better rephrase as medical, antiseptic.. applications.
Response: The language has been changed here in lines 44-45.
- Line 53, while triterpenoids are indeed synthesized via the MVA pathway, the final products are, at ambient temperature, not volatile and hence, cannot contribute to aroma. The other terpenoid subclasses named here, are volatile.
Respone: This error has been modified in the text.
- Line 55, While indeed terpene synthases (please correct synthetases) are key enzymes that determine the terpenoid profile, both the MVA and MEP pathway produce what is considered to be the direct substrates for TPA, but do not include them.
Respone: It has been corrected to terpene synthase (55 lines) ; here we just want to express that TPS is a key enzyme in the synthesis of terpenes, rather than being included in the MVA and MEP pathways, and the substrates produced by the MVA and MEP pathways that can be used to synthesize terpenes are not described in detail in the introduction. The expression has been changed here (lines 55).
- Line 60, ‘other metabolomic pathways’ is too general and can relate to any pathway not specific for benzenoid biosynthesis.
Respone: Changes have been made here (lines 59-60)
- Line 72, not clear what is meant with ‘the atypical synthetic pathways’.
Respone: The meaning of 'the atypical synthetic pathways' is 'the noncanonical biosynthesis pathways', as the literature'[18] My way : noncanonical biosynthesis pathways for plant volatiles. 'Corrected in text (line 72).
- General suggestion: do not use the term ‘also’ too many times
Respone: The term the term 'also' has been completely changed.
- Line 81, Yucatan, Cachuca and CNPH are not volatile compounds, such as is now referred to in this sentence.
Respone: Yucatan, Cachuca and CNPH4080 are pepper varieties, the description of which has been changed in the text (80-81line).
- Line 83, ‘but’ suggests a contradiction. However, this sentence is actually giving more details on the timing of emission.
Respone: 'but ' has been changed to 'Moreover' (line 82).
- Line 92, and Line 103. The authors intend to profile a ‘ characteristic local variety’, but the selection of the other three varieties was mot motivated.
Respone: We describe ' Hainan huangdenglong pepper (HDL1) ' here because we want to express the importance of this study from a specific perspective. In addition, as a unique treasured local variety in Hainan Province, its fruit smells fragrant, and the processed product Hainan huangdenglong pepper sauce is the most popular and most promising characteristic agricultural product in Hainan Province. We hope to clarify the main components, content and changes of its volatile aroma components, and to understand whether its volatile aroma substances are different from other varieties. At the same time, most of the pepper varieties planted in China belong to Capsicum chinense L.and Capsicum annuum L. Therefore, we selected three other pepper varieties from the aspects of genetic relationship, cultivated species and varieties. 'HDL2' (a hybrid variety with different fruit shape and HDL1) is a variety with close genetic relationship with 'HDL1'. 'GJ' belongs to the same cultivated species as 'HDL1', a more popular Capsicum annuum variety 'CTJ' (Capsicum annuum L.). This is briefly described in the material method (lines 585-593 line).
Results
- Line 122, 123 what is meant with ‘the kinds’ ? The biochemical classes, or specific individual metabolites? ‘A relative content of total volatiles’ is contractionary. I guess here is meant: the relative content of specific compounds in the total blend showed significant differences. Please clarify.
Respone: "The kinds of volatile compounds" refers to "the types of volatile compounds", which are specific individual metabolites. In the text, 'A relative content of total volatiles' has been corrected to 'the total relative content of volatiles' ( 125-126 lines ).
- Line 137, please specify what is meant with: the first category of volatiles.
Respone: 'the first category of volatile' refers to 'the highest content of volatiles', which has been changed in this paper (line 139).
- Figure 1 In panelA, the explanation of the Y-axis is missing in the figure. The use of both *** and P-values, is confusing. Better use the *** in the figure and explain in the legend what this stand for, ***, P<0.001 etc. What is meant with ‘groups’ on the X-axis? Are these the varieties? And in panel B, what is a, b, c in this grouping? Please adapt the legend accordingly. In legend of biochemical subclasses, all terms are written starting with a Capital, except for ketones.
Respone: The annotations in Figure 1 have been changed and simplified as shown in Figure 2a; 'groups on the X-axis' originally represented 'varieties of pepper', which has been changed in Fig.2a; a, b, and c represent three developmental stages, respectively, explained in Figure 1 (lines 110-113); the corresponding legend has been adjusted, see 'Figure 2b'.
- Line 161, differences were not analysed by relative content, this is not a method of analysis. What is most likely meant here: Differences in varieties and throughout development were analysed using the relative content of each of the metabolites.
Respone: 'The difference of VACs in different varieties and developin stages was analyzed by relative content.' means 'Differences in varieties and throughout development were analyzed using the relative content of each of the metabolites' has been modified in this paper (164-165 Lines).
- Line 164, 165. ‘excellent reproducibility’ suggests a technical replicate, not a biological one. Please clarify what exactly was the experimental set-up.
Respone: What we want to express here is that the three biological replicates of the same variety at the same developmental stage are well clustered together, indicating that the data is reliable and has been changed in the article (166-167 lines ).
- Figure 2, more detailed information should be given in the legend, including the chosen algorithm to come to this heatmap.
Respone: More detailed legends have been added here (lines 176-177).
- Also in the accompanying text, I suggest to include a description of the main results, including that metabolites derived from specific biochemical pathways do not show obvious clustering, while a clearlyseparation of green stages versus the breaking and maturation stage is visible in the abundances of specific metabolites.
Respone: Changed in text (lines 169-172)
- Line 185 ‘were chosen’, ‘three stages of development’.
Respone: Thank you for your correction, which has been revised in this article.
- Line 174-223. I’m a bit surprised by the motivation of the authors to first do PLS-DA analysis and include these in the supplemental data, including a list of discriminant compounds with proper significance (high VIP, low P values, high log2FC), while the main manuscript shows PCA plots, for which no quantitative data for the identified marker metabolites are given. While both analysis are correct in itself, I am curious to the motivation of the authors.
Respone: PLS-DA analysis is a routine analysis method for analyzing differential metabolites, which has been used in many literatures. PLS-DA analysis is used to screen the differential metabolites between specific comparison groups, and there are many comparison groups in pairwise comparison. However, we want to compare the differences in metabolites between these comparison groups ( CTJ-b VS CTJ-a, CTJ-c VS CTJ-a, GJ-b VS CJ-a GJ-c VS GJ-a, HDL1-b VS HDL1-a, HDL1-c VS HDL1-a, HDL2-b VS HDL2-a, HDL2-c VS HDL2-a, GJ-c VS CTJ-c, HDL1-c VS CTJ-c, HDL2-c VS CTJ-c ) ( Table S2 ). For other comparison groups ( CTJ-a VS GJ-b, GJ-c VS HDL1-b, HDL1-b VS CTJ-a and etc. ), there was no comparative significance in this study. Therefore, we used PLS-DA method to analyze the differential metabolites in the comparison group that needed to be analyzed in this study. The VIP value is the weight value of metabolites in different comparison groups based on PLS-DA analysis. When VIP > 1, it represents that the metabolite is important in all metabolites and has analytical significance. Therefore, based on VIP > 1, p < 0.05, and |log2FC|≥1, we screened 70 metabolites with analytical significance and significant differences in the established PLS-DA model.
The PCA plots below are based on these 70 ( rather than 149 ) differential metabolites to analyze the marker aroma compounds of pepper fruits at different stages, as in the literature "24.Liu, R., Xiong, K., Chao-Luo, Y... Changes in volatile compounds of a native Chinese chilli pepper ( Capsicum frutescens var ) during ripening. International journal of food science & technology. 2009,44 ( 12 ). 2470-2475. https://doi.org/10.1111/j.1365-2621.2009.02039.x" as described.
- Furthermore, the fact that specific metabolites are deviating from a specific comparison in the PCA plots, is not perse an indicator for being a ‘marker metabolite’ Also, there is no motivation for showing PC1, PC2, and/or PC3 in various combinations. Hence, this seems a bit arbitrary cherry picking. The legends of Figure 3, should also be more neutral. Specifically: what is meant with ‘young stage’ in figure3b right panel?
Respone: As described in 'Literature 24', we only hope to preliminarily screen ' marker metabolite ', or the most different metabolite, through this analysis method. In fact, the metabolites obtained by PCA analysis in 'literature 24' are called 'marker metabolite'. We believe that this is a preliminary screening, which can reflect the importance of different metabolites under the same conditions. Such metabolites can reflect the overall characteristics of the sample in a certain situation, so it is called 'marker metabolite'. More detailed 'marker metabolite' needs further experiments.
- Line 247/248. The fact that many genes are down-regulated is not perse explanatory for their importance in formation and release of volatile metabolites. This is an overall transcriptome, and includes genes that are related to various processes not related to formation of aroma compounds, but most likely with other aspects of fruit ripening, including many processes that are shuttled down in the final developmental stage. In this stage of the description of the transcriptome analysis, no relations with any biochemical gene expression can be made, and most specifically not with volatile formation at all. Actually, the authors nicely describe such relations in the following section, where they compare go-annotations in various comparisons. Here, it appears that indeed many genes putatively related to the formation of volatile compounds are enriched.
Respone: Here, DEGs are down-regulated more than up-regulated. We believe that down-regulated differential genes may be more closely related to the metabolic activity of pepper. Therefore, we want to express that the relationship between down-regulated differential genes and the synthesis of aroma volatiles may be more important than up-regulated DEGs. This has been changed in the text (lines 242-243).
- Legend of figure 4 should be more informative, such as describe the 36 samples, in a bit more detail: genotypes / developmental stage…
Respone: The description of 36 samples is explained in the legend of Figure 1.
- Line 259, what is meant with ‘the effect of genes (which can be read as treatment?) at transcriptional level?
Respone: has been changed to 'the change of genes' (252 lines)
- Figure 5, Please be more detailed in legend descitpion: differentially expressed genes as result of ….? / Venn diagrams indicating DEGs that are overlapping in specific comparisons. Why are some terms in panels g, h, m, highlighted? What is the reason to name the last subpanel ‘m’ instead of ‘i’? This suggest that a draft version of the figure is shown here. Please check.
Respone: In the text has been changed to 'Functional enrichment analysis of DEGs in different developmental stages.' (285 line) ; the prominent pathways in g, h, i are related to the metabolism of volatile aroma substances, in order to facilitate readers to find ; here 'm' has been changed to 'i', as shown in Figure 6i
- Line 303, 90/35 what? Genes, metabolites, other?
Respone: refers to the gene, has been changed in the text description (293-294 lines)
- Figure 6, what is the colour coding bar indicating? Expression level, Z-scores other?
Respone: The color coding table is the expression level of the gene, which has been annotated in the text (lines 332-333)
- Line 359, differential related to variety or developmental stage?
Respone: The problem here I did not find in the text.
- Line 411, what is used to identify whether a specific gene encodes for a transcription factor. Please indicate (but perhaps this already done in the method section)
Respone: The determination of TFs is based on the annotation results of the original transcriptome data.
- Line 415, Furthermore (typo) / formulation of this sentence is unclear, please adapt.
Respone: This has been modified in the text ( 405-408 lines )
- Legend of figure 8 is unclear: what is meant with ….at different spreading time points….; what are the different colours depicted in panel b indicating? Tekst in hexagons is not readable, is there a accompanying supplemental file, in which more details are presented?
Respone: The expression here is wrong and has been changed in the text ( 412-413 lines ) ; the same color in Fig.9b represents a TFs family ; hexagon represents the structural genes related to the biosynthesis pathway of aromatic substances ; the ID number of these genes is marked in the figure, and there is no other information, which can be found in NCBI through the ID number of the gene. In this part, we performed a correlation analysis based on the differential TFs obtained above and the key DEGs that regulate the synthesis of volatiles (|r| > 0.9 , P-value < 0.05).
- In panel a of figure 9, colors are mentioned, but it is not clear to what this refers (most likely figure 8 panelb?). Panel b & c, indicate which subclass of TFs is meant here, not just give colors, what are the bars indicating?? Please inducate what is specifically / additionally shown in panels d & e.
Respone: Figure 10 (Original Figure 9) is a joint analysis of differential genes and differential metabolites based on WGCNA. It has nothing to do with Figure 9 (Original Figure 8). The color in the picture only represents different modules (this color represents only a name here, for the convenience of readers). Panel b & c is the gene expression heat map of the two metabolites that we selected, and panels d & e are the correlation network maps corresponding to some genes with high correlation in Panel b & c.Generally, we believe that in the network map, the more closely linked genes may be more important. Annotation information has been added to panels d & e (446-447).
- Line 603, is there additional information on the HDL2 hybrid?, what are the parents of this hybrid, is one of them HDL1? Is there any more detailed (genebank) description of the specific varieties used? At least, geographical indicators, or were fruits just bought at local markets?
Respone: “HDL2” Pepper is a hybrid variety purchased from the market. It is an exotic Huangdenglong Pepper hybrid currently being vigorously promoted in Hainan Province.(588-589 Lines)
- Line 606, what were the conditions in the plant growing chambers?
Respone: Plant growth chamber information has been added to the text (594-595 lines)
- Line 617-626.
Please provide more details on the metabolome analysis. Has the risk of losing specific (very volatile) metabolites due to the freeze-drying procedure being validated? Please provide details (reference, in supplemental). What was the concentration of the ethanol (70%, 96%, other) and how much volaum was added. What is the volume and the provider of the headspace bottles used. Why were samples incubated at such a high temperature, is this representative for smelling pepper fruits? I would expect either an ambient 20-25C or an oral-environment 37C incubation. Please motify.
Respone: The 'freeze-dried 'expressed in this paper refers to the solid state of fresh pepper fruits in liquid nitrogen, which is better ground into powder to improve extraction efficiency. In fact, we use fresh fruit samples after thawing. In this article we did not express clearly, has made the change (606-608 lines). Ethanol concentration and volume (610 lines) were added in the paper; headspace bottles supplier added (611 lines) ; 60℃ extraction is based on the methods in references 52 and 53, which can improve the extraction efficiency. References (615 lines) have been added to the article.
- Line 624, unclear formuation. I suspect that a PDMS fiber was inserted into the headspace of the extracts. Were indeed individual samples incubated for 60 min? Would you not expect a high amount of degradation products due to the extraction and incubation, instead of representative for the composition of the fruits themselves? Has this method been validated? In my lab, we usually incubate SPME fibers for 10-20 min max to avoid that we actually trap other compound than intended.
Respone: The extraction temperature and time of this process are the references "52. Patel K, Ruiz C, Calderon R. Characterisation of volatile profiles in 50 native Peruvian chili pepper using solid phase microextraction-gas chromatography mass spectrometry (SPME-GCMS) [J]. 2016, 89, 471-475. https://doi.org/10.1016/j.foodres.2016.08.023" and "53, Mazida M M, Salleh M M. Osman H. Analysis of volatile aroma compounds of fresh chilli (Capsicum annuum) during stages of maturity using solid phase microextraction (SPME) [J].Journal of Food Composition and Analysis, 2005. 18 (5), 427-437. https://doi.org/10.1016/j.jfca.2004.02.001"
- Line 640, where QC samples actually included in the data analysis, are these shown in the supplementary data?
Respone: QC samples are included in the data analysis, see Figure S2.
- Line 644, what is meant with substances? Better use ‘mass spectra’ and metabolite indentation.
Response: 'substances 'represents 'VACs ', which has been changed (633 Lines).
- Section 5, Conclusion. Actually, I would have expected to read this concluding section at eh end of the discussion section.
Response: The conclusion is placed at the end of the article because the structure of the article is sorted according to the IGMS journal manuscript layout requirements.
Round 2
Reviewer 1 Report
Why is the author response to comments written in Chinese?
Author Response
I 'm very sorry, I don 't know the problem of Chinese, I resubmitted in the form of documents.Please see the attachment

Reviewer 2 Report
The author has made some changes to the paper, but it is only a statement of tinkering, not enough to be published. The author needs to improve comprehensively from the overall idea of manuscript writing, the specific chart, table arrangement and analysis.
What data is most important that the reader must have easy access to the information to put into the text., and which are attached?
For example, in Figure 1 and 5, etc., there are still many problems in data analysis. Significance analysis only needs to mark significant differences, instead of writing p< 0.001, 0.0001.
p < 0.001 and p < 0.0001 cannot be used to determine who is more significant.
Of course, the problems in this article are not limited to the above.
Therefore, , it is not enough to be published as present form.
Author Response
Thank you again for your valuable suggestions for this article. We have made some modifications to this article based on your review recommendations and the recommendations of other reviewers, including graphics production, data analysis, results elaboration, and chart sorting. In this paper, we put in a statistical table of all metabolites, and made a more easily distinguishable significance of Figure 2a. In addition, we also re-combed and re-interpreted the results analysis section.
Here we also need to explain to you, some of the table we did not put in the text, not that it is not important, but the amount of data in these tables is too much, will occupy too much text space, as the text of the table 1, in addition to the amount of data analyzed in this paper and put in the manuscript of the number of charts, therefore, some important charts can not be fully put in the text, can only be provided to the reader in the form of attachments.
The manuscript submitted this time is a revised version based on the suggestions of you and other reviewers. If you have other specific suggestions and doubts about our manuscript, please inform us in time and we will make adjustments in time.
Looking forward to your next communication!
Round 3
Reviewer 1 Report
This manuscript can be accepted in present form.
Reviewer 2 Report
The logic of the language is a little clearer. For example, "we found significant quantitative differences in VACs from several Chinese spicy pepper and one India pepper", but it should be said that Chinese spicy pepper and one India pepper have different VACs content and types, and the main reason for this difference are esters, especially in Hainan Huangdenglong .